

**A three year time-series of volatile organic iodocarbons in Bedford Basin, Nova Scotia :**
**a Northwestern Atlantic fjord.**
*Qiang Shi[1], Douglas Wallace[1]*
[1]Department Oceanography, Dalhousie University, Halifax, Canada
Email: qshi@dal.ca
**Abstract:**
We report weekly observations of volatile organic iodocarbons ($CH_3I$, $CH_2ClI$ and $CH_2I_2$) over the
time-period May 2015 to December 2017 from 4 depths in Bedford Basin, a coastal fjord (70m
deep) on the Atlantic coast of Canada. The fjord is subject to winter-time mixing, seasonal
stratification and bloom dynamics, subsurface oxygen depletion, local input of freshwater and
occasional intrusions of higher density water from the adjacent continental shelf. Near-surface
concentrations showed strong seasonal and sub-seasonal variability which is compared with other
coastal time-series. Relationships with other properties lead to the hypothesis that near-surface
iodocarbon production is linked to reduction of iodate to iodide under conditions of post-bloom
phytoplankton senescence, nutrient stress or viral lysis following seasonal disappearance of
nutrients. The vertical variation of $CH_2I_2$ and $CH_2ClI$ within the upper 10m is consistent with rapid
photolysis of $CH_2I_2$.
Average annual sea-to-air fluxes (62 nmol $m^{-2}$ $day^{-1}$) of total volatile organic iodine were slightly
higher than observed in other coastal and shelf time-series and polyiodinated compounds
contributed 85% of the total flux. Fluxes were subject to strong interannual variability (2-3X) as a
result, mainly, of wind-speed variability. Near-surface net production of $CH_3I$ averaged 1.0 pmol
$L^{-1}$ $day^{-1}$ and was similar to rates in the English Channel but an order of magnitude higher than in
shallow waters of the Kiel Fjord, Germany, possibly due to higher microbial degradation in the
latter.





The near-bottom (60 m) time-series showed evidence for CH$_3$I production associated with organic
matter degradation, and a possible "switch" from production of CH$_3$I via an alkylation pathway to
production of CH$_2$I$_2$ by a haloform-type reaction. Near-bottom CH$_3$I production varied strongly
between years but was generally ca. 20 times lower than near-surface production. Post-intrusion
decreases of iodocarbons at 60m suggested removal timescales of 14, 65 and 70 days for CH$_3$I,
CH$_2$I$_2$ and CH$_2$ClI respectively.
**Keywords:** Iodocarbons, iodomethane, chloroiodomethane, diiodomethane, air-sea flux, time-
series
1.   Introduction
Volatile organic iodocarbons (VOIs) such as methyl iodide (CH$_3$I), chloroiodomethane (CH$_2$ClI)
and diiodomethane (CH$_2$I$_2$) have a predominantly oceanic source and supply a significant amount
of iodine to the atmosphere (see review by Saiz-Lopez and Von Glasow, 2012). These gases, also
referred to as VSLS (very short-lived source gases) due to their reactivity and short atmospheric
lifetimes, have been implicated in supporting catalytic ozone destruction in the troposphere (Davis
et al., 1996; McFiggans et al., 2000a) and, potentially in the lower stratosphere (Solomon et al.,
1994) as well as aerosol formation in the marine boundary layer (Mcfiggans et al., 2004;
McFiggans et al., 2000b; O'Dowd et al., 2002). Recent modelling of atmospheric reactive iodine
(IO$x$ = IO + I) as well as experimental studies (Carpenter et al., 2013; Jones et al., 2010; Mahajan
et al., 2010) suggest that the supply of volatile organoiodine represents <50% of the total sea-to-
air delivery of reactive iodine, with most being supplied in the form of HOI and I$_2$. Nevertheless,
the potential for localized higher emissions coupled with their relatively long lifetimes (compared
to I$_2$ and HOI) allows the organic compounds to be a significant source of iodine to the free



troposphere and even, potentially, to the lower stratosphere in certain regions (Tegtmeier et al.,
2013). Further, *Mahajan et. al* (2012) noted a strong correlation of $IO_x$ and $CH_3I$ suggesting that
the sources of $CH_3I$ and the shorter-lived precursors of $IO_x$ are closely related or depend on similar
variables.
$CH_3I$ is the most abundant VOI species in the atmosphere (Yokouchi et al., 2011) because of its
longer lifetime (days) compared to $CH_2ClI$ (minutes) and $CH_2I_2$ (hours) (Moessinger et al., 1998;
Rattigan et al., 1997). However, the total supply of organically-bound iodine to the atmosphere is
several times larger than the $CH_3I$ supply alone (Carpenter et al., 2014) with the bulk of the
remainder transported in the form of $CH_2I_2$ and $CH_2ClI$. Despite considerable attention on the
oceanic distribution and sea-to-air flux of these compounds, in particular $CH_3I$ (Ziska et al., 2013),
it is not yet possible to apportion oceanic production of these compounds, unequivocally, to
specific mechanisms. Even for $CH_3I$, controversy remains, for example, as to the relative
importance of direct "biological" or "photochemical" production pathways with experimental
evidence reported for both, and correlation analysis generally being inconclusive, in part because
of the "snapshot" nature of most studies (Stemmler et al., 2014). Comparisons of models to
observed distributions have also proven ambiguous, with localized studies suggesting
predominance of a biological production pathway (Stemmler et al., 2013) but a global analysis
emphasising photochemical production as the dominant mechanism. This diversity of views has
been maintained through a variety of experimental studies (Amachi et al., 2001; Brownell et al.,
2010; Hughes et al., 2011; Manley and delaCuesta, 1997; Moore and Tokarczyk, 1993; Moore and
Zafiriou, 1994; Richter and Wallace, 2004; Shi et al., 2014a; Smythe-Wright et al., 2006).
For compounds other than $CH_3I$, similar uncertainty exists concerning production pathways, but
with fewer underlying studies. Laboratory experiments have shown that the presence of dissolved





iodide and dissolved organic matter can lead to production of these compounds in the dark
(Martino et al., 2009). *Fuse et al.* (2003) and *Martino et al.* (2005) observed that $CH_2ClI$ could be
produced by photolysis of $CH_2I_2$ in artificial and natural seawater. However detailed mechanisms
and, especially, their relative importance in the field remain unclear.
Time-series observation can reveal processes and controlling factors underlying production and
loss of iodocarbons in the ocean and provide data for testing hypotheses and/or models. However,
only a very few long-term, time series observations of iodocarbons have been reported to date, all
from coastal water. *Klick* (1992) reported 13 months of weekly measurements of $CH_2I_2$ and $CH_2ClI$
from very shallow (3.5m) water in the Kattegat at the Swedish coast. *Orlikowska and Schulz-Bull*
(2009) reported a year of weekly data for $CH_2ClI$, $CH_2I_2$, $CH_3I$ and $C_2H_5I$ from a nearshore (3m
depth) site in the Baltic Sea. *Archer et al.* (2007) reported a seasonal study of $CH_2ClI$, $CH_2I_2$, $CH_3I$,
$C_2H_5I$, and $CH_2BrI$ measured weekly at 4 depths (0-50m) in the western English Channel from
July 2002 to April 2004. *Shi et al.* (2014b) reported on the seasonal cycle of $CH_3I$ from surface
waters of the Kiel Fjord: a shallow (14 m), brackish water body in northern Germany, which was
sampled weekly for 2 years. *Shimizu et. al* (2017) presented a time-series of vertical profiles (0-
90m) of $CH_2I_2$, $CH_2ClI$, $CH_3I$, and $C_2H_5I$ from the centre of Funka Bay, Japan, which were
measured every 2-4 weeks from March 2012 to December 2014.
Here, we report weekly observations of $CH_3I$, $CH_2ClI$ and $CH_2I_2$ made over the time-period May
2015 to December 2017 at 4 depths (0-60m) in Bedford Basin: a coastal fjord on the east coast of
Canada. We report seasonal to interannual variability of the observed concentrations at different
depths in the water column and compare our results with the other time-series. We report daily
average fluxes to the atmosphere and use a simple, time-varying mass-balance model for near-
surface waters to estimate production rates and their variability. We discuss the observed



variability of both concentrations and production rates in the light of earlier studies, potentially
correlated variables and suggested production pathways.
2. Method
Time-series measurements of VOIs were carried out in the Bedford Basin (44.69 °N, -63.63 °E)
near Halifax, Canada. Bedford Basin is an 8 km long, 17 km$^2$ fjord with a maximum depth of 71m
and a total volume of 500 km$^3$. The Bedford Basin is connected with continental shelf waters of
the Atlantic Ocean through "the Narrows" (a ca. 300 m wide and 20 m deep passage (fig. 1). The
Basin receives freshwater primarily from the Sackville River at its northwestern end, with a total
average freshwater input of 5.41 m$^3$ s$^{-1}$ (Buckley and Winters, 1992). The average near surface
salinity within the Basin is 29 which can be compared with salinities of >30 over the adjacent
Scotian Shelf. There are only relatively small horizontal gradients of near-surface salinity within
the Bedford Basin itself (typically < 2 difference from close to the Sackville River mouth to the
Narrows).
Time series observations of physical, chemical and biological parameters have been recorded since
1992 (Li, 1998). Our halocarbon samples were collected weekly, in the center of the Bedford Basin,
at its deepest point (Figure 1), between May 2015 and January 2018. Samples were collected with
10-L Niskin bottles attached to a rosette sampler at 1, 5, 10 and 60 m (10 m samples were collected
biweekly from May to September 2015). The upper three water samples covered the majority of
the euphotic zone. The 60m water sample was from typically stagnant, near-bottom water which
is renewed by vertical mixing events in late winter, and by occasional intrusions of higher-salinity
continental shelf water in both summer and winter. Chlorophyll *a* (Chl*a*), dissolved oxygen, and
nutrients were measured weekly at the 4 depths as part of the Bedford Basin Monitoring Program



(Details can be found in website: http://www.bio-iob.gc.ca/science/monitoring-monitorage/bbmp-
pobb/bbmp-pobb-en.php). In addition to the Niskin bottle sampling, vertically continuous
measurements of temperature, salinity, dissolved oxygen and Chl *a* properties were measured with
a CTD mounted on the rosette. Additional information concerning the measurements of supporting
physical and biological parameters can be found in the paper by *Burt el al.* (2013).
The concentrations of iodomethane ($CH_3I$), chloroiodomethane ($CH_2ClI$) and diiodomethane
($CH_2I_2$) reported here, as well as of a number of other halocarbons (data not shown), were
measured using purge & trap gas chromatography with detection by both mass spectrometry (MS)
and electron capture (ECD). All measurements were made using an Agilent Technologies gas
chromatograph (GC 7890B), equipped with a capillary column (RTX-VGC; 60 m; 1.4 μm coating,
column diameter: 0.25 mm; helium carrier gas 0.5 ml min$^{-1}$), together with an automated purge
and trap system equipped with an autosampler (VSP4000 of IMT, Vohenstrauss, Germany). The
GC column was temperature programmed as follows: initial temperature 50 °C for 6 minutes, then
ramped to 150 °C at 6 °C min$^{-1}$; ramped to 200 °C at 10 °C min$^{-1}$. Water samples (10 ml) were
stored in 20 ml vials equipped with an ultra-low-bleed septum, prior to purging with helium (20
ml min$^{-1}$ for 18 mins). Every sample was analysed in triplicate. The standard deviation of triplicate
measurements (integrated peak area) was <10 % for $CH_3I$, <15 % for $CH_2ClI$ and <20 % for $CH_2I_2$.
Calibration of the GC system for $CH_3I$, $CH_2ClI$ and $CH_2I_2$ was performed using permeation tubes
(VICI, Houston, TX, USA) which were maintained at a constant temperature of 23 °C and weighed
every 2 weeks. Dilutions of the permeation tube effluent were made in ultra-high-purity $N_2$
(>99.995 %) with flow rates of 50 to 700 ml min$^{-1}$, and samples were injected into the purge and
trap system (VSP) through a 140 μl loop. Standard deviation of the peak area during these



calibration runs was <5 % for CH$_3$I and CH$_2$ClI and <15 % for CH$_2$I$_2$. Overall the calibration
response varied by less than 15 % over the entire sampling period.
Throughout the paper, seasons are defined as follows: summer is June through August; fall is
September through November; winter is December through February and spring is March to May.
3.   Results and discussion
3.1 Environmental Variables from the Bedford Basin
Figures 2 and 3 depict time-depth plots of the variation of sea surface temperature (SST), salinity
(SSS), dissolved oxygen, total dissolved inorganic nitrogen (DIN), and fluorescence in Bedford
Basin over the period of the VOIs sampling. The vertical profiles of temperature and other
properties are well-mixed from top to bottom in late winter (Feb-Mar) as a result of wind-mixing
and convection (Li, 2001). Temperature is marked by strong seasonality to depths of <30 m. Near-
surface temperatures start to rise above winter values of 4 °C, and stratified conditions develop,
around early April with temperatures reaching ca. 21 °C by the end of August (fig. 3a). Typically,
strongest irradiance (data was downloaded from the CERES FLASHFLUX system:
https://power.larc.nasa.gov/cgi-bin/hirestimeser.cgi) occurs in June and July, and highest water
temperatures are observed in August (see also fig. 7).
Salinity ranges from 23 to 31 through the entire water column, with the lowest salinities occurring
very close to the surface (fig. 2b, c and fig. 3b). The near-surface stratification varied both
seasonally and between years, primarily in association with variability of precipitation and the
discharge of the Sackville River (source: Environment and Climate Change Canada;
http://climate.weather.gc.ca/historical_data/search_historic_data_e.html).    For    example,    the
salinity at 1m was 1psu lower than at 5m during much of the summer of 2015 (June to September)
and summer 2017 (June to August). In summer 2016, however, the salinity at 1m was close to that



at 5m (difference < 0.3) (fig. 2b). The average precipitation in Bedford Basin was 27.6 mm week$^{-}$
$^{1}$ in summer 2015 and was 16 mm week$^{-1}$ in summer 2016 (fig. 2d). Occasional intrusions of more
dense water from the Scotian Shelf, results in increased salinity, especially of bottom waters. The
intrusions are irregular and tend to occur a few times per year, for instance in early November
2016 at which time the salinity of bottom water increased from to 30.9 to 31.3.
The dissolved oxygen time-series (fig. 3c) shows the effect of temperature-dependent solubility
variations in surface waters as well as intrusions and late-winter vertical mixing in deeper water.
In surface water the highest $O_2$ concentrations occurred between March and April every year in
association with lowest SST. The vertical gradient of $O_2$ concentration was, generally, smallest
towards the end of April as a result of vertical mixing. Sub-surface $O_2$ concentrations (>30 m)
generally decreased in summer due to respiration, with occasional interruptions of this $O_2$ decline
(e.g. November 2016) as a consequence of shelf-water intrusions which brought sudden increases
in $O_2$ levels.
The seasonal variations of chlorophyll *a* concentration in surface water are plotted in figure 2a.
Typically, two blooms (spring and autumn) occur in surface water. For example, in 2016,
chlorophyll *a* increased rapidly from March to April (from 5 to 26 μg L$^{-1}$), and from September to
October (from 10 to 28 μg L$^{-1}$). The vertical variation of chlorophyll *a* (as determined from
fluorescence measured on the CTD, see fig. 3d) reached 12 μg L$^{-1}$ during the bloom period. Sub-
surface (20- 40 m) fluorescence-derived chlorophyll *a* dropped down to 4 μg L$^{-1}$. In the near-
bottom water chlorophyll *a* ranged between 0 and 2 μg L$^{-1}$ during the whole year and varied only
slightly.
The seasonal variation of dissolved inorganic nitrogen (DIN=$NH_4^+$ + $NO_2^-$ + $NO_3^-$) in surface
water is plotted in figure 2b. In winter, when chorophyll *a* levels are very low due to light limitation,



DIN concentrations reach ca. 12 μmol L$^{-1}$ but are drawn down to low levels (< 1 μmol L$^{-1}$) after
the spring bloom. Summertime chlorophyll *a* levels are moderate but variable (ca. 3 to 10 μg L$^{-1}$),
likely reflecting continuing nutrient input (e.g. from runoff and/or sewage treatment plants).
3.2 Variations of Iodocarbons Concentrations in Bedford Basin
Iodocarbon concentrations in surface water (1, 5 and 10 m) showed strong seasonality, with lowest
concentrations from December through May (1.2 pmol L$^{-1}$ for CH$_3$I; 1.3 pmol L$^{-1}$ for CH$_2$ClI and
0.3 pmol L$^{-1}$ for CH$_2$I$_2$). Concentrations start to increase in late May/ June, reaching levels as high
as 45 pmol L$^{-1}$ for CH$_3$I; 160 pmol L$^{-1}$ for CH$_2$ClI and ca. 80 pmol L$^{-1}$ for CH$_2$I$_2$ (with a single
peak of 500.5 pmol L$^{-1}$; fig. 4). Near-surface, summertime concentrations of all three compounds
were characterized by a broad seasonal peak of 6-7 months duration (or shorter for CH$_2$I$_2$), on top
of which were superimposed ca. 3-4 peaks of shorter (4-5 week) duration. The number, amplitude
and timing of these peaks varied amongst the three compounds with CH$_3$I, notably, showing only
one large peak in 2016 and four during the other two years of the time-series (fig. 4a).
Concentrations at 60 m were almost always lower, and much less variable, ranging over the year
from 1 to 9 pmol L$^{-1}$ for CH$_3$I (except the Fall/Winter 2015-2016, see below), 1 to 6 pmol L$^{-1}$ for
CH$_2$ClI and 0.4 to 18 pmol L$^{-1}$ for CH$_2$I$_2$ (fig. 4d) respectively. Hence, the bottom water (60 m)
concentrations of CH$_2$I$_2$ and CH$_2$ClI were always much lower than in near-surface waters
throughout the summers. The surface to deep concentration difference was smallest for CH$_3$I and
showed interannual variability. Notably bottom water concentrations reached 26 pmol L$^{-1}$ and were
even higher than in contemporary surface waters from September 2015 to March 2016 (fig. 4d).
Missing from the bottom water time-series, were the ca. 1 month duration variations seen in
summertime surface water.



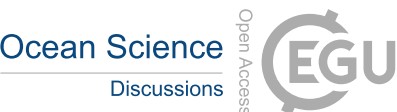

Inter-relations between the iodocarbons in surface seawater were examined with linear regression
of both weekly and monthly-averaged concentrations. The resulting correlations are shown in table
1. Using weekly data, significant correlations (i.e. $p < 0.05$) were found between [$CH_3I$] and
[$CH_2ClI$] at 1, 5 and 10 m depths with the strongest correlation (0.7) at 10m. The only other
significant correlation was between $CH_2I_2$ and $CH_2ClI$ at 5 m. Use of monthly averaged values
gave stronger correlations. Once again, the significant correlations were between $CH_3I$ and $CH_2ClI$
(at 1, 5 and 10m depth) as well as between $CH_2ClI$ and $CH_2I_2$ at 5 and 10m depth. Table 2 also
presents the correlations of iodocarbon concentrations with potentially related variables (discussed
in section 4.3).
Generally, the concentration of $CH_2I_2$ was higher than that of $CH_2ClI$. The average ratio of
$CH_2I_2/CH_2ClI$ within the top 10m of the water column over the summer months was 1.4. However,
this ratio was significantly lower at 1m depth (average of 0.6) and increased with depth (1.5 at 5
m and 2.2 at 10 m, reaching values as high as 2.7 at 60 m).
3.3 Sea-to-Air Flux
Using the concentrations of $CH_3I$, $CH_2ClI$ and $CH_2I_2$ at 1m depth (Figure 4a) we estimated the
sea-to-air flux of VOIs (F) using the following equations:

$$Flux = K\left(C_{aqu} - C_{air} \times H\right)$$

$$K = \left(\frac{S_c}{660}\right)^{-0.5} \left(0.222 \cdot u_{10}{}^2 + 0.333 \cdot u_{10}\right)$$

For the calculation of the sea-to-air flux (F) of all three compounds, we assumed that the
concentration in the atmosphere ($C_{air}$) was negligible. *Rasmussen et al.* (1982) reported an average
atmospheric mixing ratio of $CH_3I$ of ca. 1 pptv for Cape Meares (45 °N) and *Yokouchi et al.* (2008)
presented a mean concentration of 0.98 pptv for Cape Ochiishi (43.2 °N), with both sites sharing



a similar latitude to our sampling location (44.69 °N). If a mixing ratio of 1pptv is applied to our
flux calculations, the total annual flux would be reduced by only 5 %. The atmospheric mixing
ratios of $CH_2ClI$ and $CH_2I_2$ are lower than those of $CH_3I$ (reviewed by Carpenter, 2003) and, hence,
any overestimation of the fluxes of these compounds will certainly be negligible (Archer et al.,

5   2007).

We calculated the fluxes of the three compounds using the transfer velocity (K) parameterization
(Nightingale et al., 2000), where Sc is the temperature-dependent Schmidt number, as estimated
by *Groszko* (1999) and Henry's Law constants (H) were from *Moore et al.* (1995). The wind speed
data (daily averages, fig. 2e) were measured at the nearby Halifax Dockyard (fig. 1) (source:
Environment and Climate Change Canada; http://climate.weather.gc.ca/index_e.html).
The calculated daily emissions ranged from 0.3 to 78.9 nmol $m^{-2}$ $d^{-1}$ for $CH_3I$ (annual average of
8.7 nmol $m^{-2}$ $d^{-1}$), 0.3 to 203.2 nmol $m^{-2}$ $d^{-1}$ for $CH_2ClI$ (annual average of 19.6 nmol $m^{-2}$ $d^{-1}$) and
0.1 to 240.6 nmol $m^{-2}$ $d^{-1}$ for $CH_2I_2$ (annual average of 16.7 nmol $m^{-2}$ $d^{-1}$). Figure 5 presents the
combined flux of organically-bound iodine, $F_{Iorg}$, as stacked bar charts, where $F_{Iorg} = (F_{CH3I} +$
$F_{CH2ClI} + 2*F_{CH2I2})$. (Note that values plotted in Figure 5 represent daily flux estimates specific to
days on which sampling took place, and are based on the daily-average wind-speed and
concentration for those specific days.) Seasonal and annual average fluxes of the individual
compounds and of $I_{org}$ are presented in table 4. Clearly the sea-to-air flux is highest in summer and
fall and is dominated by the flux of the dihalomethanes rather than $CH_3I$.
3.4 Net Production of $CH_3I$
Making use of the air-sea flux calculations (section 3.3), we used a mass balance approach to
estimate the production rate of $CH_3I$ from the concentration time-series (see also Shi et al., 2014b).





Using the annual cycle of near-surface CH₃I concentrations (average of concentrations at 1, 5 and
10 m; Fig. 4a), we examined the mass balance of CH₃I in the top 10 m of the water column
according to:
$$\Delta C = P_{net} - L_{sea-to-air} - L_{SN2} - L_{mix}$$
where $\Delta C$ is the daily accumulation of VOIs in near-surface seawater; $P_{net}$ is the net production
rate (i.e. the net of gross production minus any additional, uncharacterized losses such as microbial
degradation); $L_{sea-to-air}$ is the sea-to-air flux (section 3.3) and $L_{SN2}$ is the 'chemical' loss due to
nucleophilic substitution of Cl⁻ for I⁻ which was calculated based on reaction kinetics (Elliott and
Rowland, 1993; Jones and Carpenter, 2007) using the corresponding temperature, salinity and
mean concentration of CH₃I. $L_{SN2}$ averaged 0.1 pmol L⁻¹ day⁻¹. $L_{mix}$ is the loss due to downward
mixing and has been shown in several studies to be negligible compared with other loss terms e.g.
(Richter and Wallace, 2004). This assumption will not be valid in winter when mixed layers deepen,
however most production of iodocarbons appears to occur during summer and fall.
The net production rate of CH₃I over the 3-year time-series is shown in figure 6. The annual
average production rate of CH₃I was 1.0 pmol L⁻¹ day⁻¹ (ranging from -1.6 to 8.5pmol L⁻¹ day⁻¹).
A significant peak of the net production rate occurred from August to September in every year.
The net production rate of CH₃I in summer and fall averaged 1.6 pmol L⁻¹ day⁻¹ and was 5 times
larger than the net production rate in winter of ca. 0.3 pmol L⁻¹ day⁻¹.
4.  Discussion
In the following we discuss the Bedford Basin data in comparison with other studies that have
reported concentrations of multiple iodocarbons and especially those that have reported time-series
covering an annual cycle (see citations in the introduction).  All of these time-series are from mid-



latitude (40-60 $^\circ$N) nearshore or continental shelf environments subject to strong seasonal
variations of light, temperature and biological productivity. There are no reported time-series of
seawater concentrations from low latitudes.
4.1 Potential influence of nearshore and /or macroalgal sources
The potential of nearshore macroalgae to cause elevated coastal iodocarbon concentrations has
been mentioned in a number of studies (Giese et al., 1999; Manley and delaCuesta, 1997; Schall
et al., 1994). We investigated this in July 2017, by sampling at 5 nearshore sites around Bedford
Basin (fig. 1) and comparing nearshore concentrations with values measured at the regular
sampling site in the center of the Basin (table 3). The results show no significant difference
between sampling locations. *Klick* (1992) also compared measurements on samples collected
directly over a rich bed of macroalgae with samples collected further away from direct contact
with macroalgae: whereas they observed significantly higher concentrations of bromocarbons in
proximity to the macroalgae, there was no difference observed for $CH_2I_2$ and $CH_2ClI$. *Shimizu et*
*al.* (2017) sampled a number of nearshore regions around Funka Bay, including rocky shores with
extensive macroalgae, and also found concentrations to be similar at both nearshore and central
Bay locations. We therefore conclude that any direct impact of macroalgae on measured
organoiodine levels is small, even in coastal regions, which lends strong support to the conclusion
by *Saiz-Lopez and Von Glasow* (2012) that macroalgae are only a minor global source of these
compounds to the atmosphere.
4.2 Concentrations and relative abundance of iodocarbon compounds



The average concentration of total volatile organic iodine $I_{org}$ (where $I_{org} = [CH_3I] + [CH_2ClI] +$
$2[CH_2I_2]$) and the relative contributions of the different compounds to $I_{org}$ from this and other
studies is shown in Figure 8. The combined concentrations of the three iodocarbons are highest
but also highly variable ($[I_{org}] = 25$ to $281$ pmol L$^{-1}$) in summertime coastal waters (loosely defined
here as within a few kms of land). Continental shelf waters have lower concentrations of $I_{org}$
averaging $32$ pmol L$^{-1}$, with open ocean waters having comparable or lower concentrations
(average $I_{org} = 17$ pmol L$^{-1}$). This distribution is contrary to the global distribution of $CH_3I$ reported
by *Ziska et. al* (2013) who noted a tendency for the open ocean to have higher concentrations than
coastal waters (their definition of "coastal" was within 1 degree latitude or longitude of land and
therefore much broader than ours). As noted by *Ziska et al.* (2013), this may reflect higher $CH_3I$
concentrations in tropical and sub-tropical open ocean waters, as their general pattern was reversed
in the Northern Hemisphere. The coastal waters depicted in Figure 8 are largely from mid-latitudes
of the Northern Hemisphere.
The relative contribution of the dihalomethanes to $I_{org}$ also varies between regions, with the ratio
of dihalomethane-I to $I_{org}$, $([CH_2ClI]+2*[CH_2I_2])/[I_{org}]$, averaging 0.71, 0.69 and 0.55 in coastal,
shelf and open ocean waters, respectively. The elevated contribution of dihalomethanes in coastal
waters is consistent with the report by *Jones et al.* (2010).
*Klick* (1992), *Jones et al.* (2010) and *Shimizu et al.* (2017) reported concentrations of volatile
organic iodine in summertime coastal waters that are comparable to, or higher than those observed
in Bedford Basin (i.e. average $I_{org}$ concentrations >100 pmol L$^{-1}$). Our results from Bedford Basin
correspond closely with the concentrations and relative contributions reported by *Shimizu et al.*
(2017) for coastal water in Funka Bay, Japan (fig. 8). In these coastal surface waters, the $CH_2I_2$
concentration and contribution was highest on average, followed by $CH_2ClI$ and the lowest was





CH$_3$I. In open ocean waters, the relative contribution of [CH$_3$I] to I$_{org}$ is higher and reaches over
50% in some cases (see fig. 8), with the contribution of CH$_2$I$_2$ generally being much lower in the
open ocean than in coastal waters.
In laboratory studies, *Fuse et al.* (2003) demonstrated that relatively large amounts of CH$_2$I$_2$ and
I$_2$ together with smaller but still significant amounts of CH$_2$ClI and CHI$_3$ can be produced,
presumably abiotically, in dark incubations of (filtered) spent culture media with suspended
bacterial cells and added [I$^-$]. The CH$_2$I$_2$/CH$_2$ClI production ratio was ∼35 and no mono-iodinated
CH$_3$I was produced in these experiments. The implication was that dissolved organic compounds
within spent media were key to production of polyiodinated compounds. In the absence of spent
culture media, additions of oxaloacetic acid also resulted in formation of CH$_2$I$_2$ and CH$_2$ClI (with
a lower ratio of CH$_2$I$_2$/CH$_2$ClI of ∼10) suggesting that organic acids may be a substrate for their
formation. The mechanistic role of the suspended bacterial cells was not clear, however they may
have supplied haloperoxidases required for oxidation of I$^-$ (see also Hill and Manley, 2009).
*Martino et. al* (2009) demonstrated that, alternatively, oxidation of dissolved iodide to I$_2$ and HOI
by reaction with ozone (e.g. Garland et al., 1980) in filtered (0.2 µm) seawater containing natural
levels of dissolved organic matter also resulted in formation of polyiodinated compounds (CH$_2$I$_2$,
CH$_2$ClI and CHI$_3$) with CH$_2$I$_2$/CH$_2$ClI production ratios ranging from 2 to 4. They suggested that
the yield of various iodocarbons depends on "the abundance and perhaps on the nature of the
organic substrate" which "can vary widely both temporally and spatially". We could not, however,
find any obvious relationship of near-surface iodocarbon concentrations with local measurements
of atmospheric ozone near Bedford Basin (results not shown).
We therefore suggest that higher levels of CH$_2$I$_2$ observed in coastal waters reflect a higher supply
rate of HOI and/or I$_2$ and/or abundance of organic precursor compounds suitable for formation of



polyiodinated compounds. Iodide is also implicated as the source of reactive iodine for the
photochemical formation of CH₃I (Moore and Zafiriou, 1994), yet *Shi et. al* (2014b) found no
positive correlation of [I⁻] with seasonal CH₃I production in the Kiel fjord. They noted however
that there may always have been sufficient iodide available to support the observed production of
pM levels of CH₃I. Given that it is the formation rate of reactive iodine (e.g. iodine atoms, HOI
and/or I₂) that is likely the key control, there is no *a priori* reason that iodocarbon production
should correlate positively with the concentration of I⁻. Indeed, formation of HOI, for example as
a result of the action of haloperoxidases or ozone, might lead to transient depletions of iodide.
Ultimately, however, iodocarbon production is likely to be controlled by the production of iodide
as this, in turn, determines the potential for production of HOI and other reactive inorganic iodine
species.
Relatively small quantities of CH₂ClI were produced in several of the experiments cited above,
yet observations in Bedford Basin show average CH₂I₂/CH₂ClI ratios of 1.4 in the top 10 m of the
water column. Production ratios in experiments clearly vary, as noted above, but laboratory studies
have also shown that photolysis of CH₂I₂ can be an important source of CH₂ClI in surface waters
with a yield of 25 % to 35 % (Jones and Carpenter, 2005; Manuela Martino et al., 2005). The
correlations between [CH₂ClI] and [CH₂I₂] in Bedford Basin (table 1) are consistent with this
photochemical transformation but may also reflect the original production ratios which could, in
turn, be substrate-dependent. In both cases however, DOM quality and quantity (possibly
associated with terrestrial supply) and/or elevated supply of I⁻ are likely to be underlying reason(s)
for the high concentrations of dihalomethanes observed in Bedford Basin and other coastal waters.
4.3 Temporal variations of iodocarbons in near-surface water





The following discussion of temporal variability is separated into consideration of seasonal, sub-
seasonal and interannual variations.
4.3.1 Seasonal Variations
All of the reported iodocarbon time-series showed strong seasonality, with minimum, sometimes
undetectable concentrations in winter, and higher concentrations in summer. Near-surface (0-10
m) concentrations of all three iodocarbons in Bedford Basin, including $CH_3I$, remained low until
mid-May to mid-June, with their subsequent increase coincident with initial warming of near-
surface waters from wintertime minimum temperatures of ca. 1-2 °C (lag < 1 month; fig. 7a). Hence
the initial appearance of all three iodocarbons occurred more than 3 months after the seasonal
increase in solar radiation, ca. 1-2 months after the Spring Bloom (fig. 7d), and after near-surface
nitrate had been drawn down to low levels (fig. 7e).
In the western English Channel (Archer et al., 2007), a gradual increase of $CH_3I$ commenced in
February, coincident with the seasonal increase in solar radiation. Summertime values remained
high, with some higher-frequency variation, and then decreased in September/October. The
increase of $CH_2ClI$ and $CH_2I_2$ started later, in April, more or less coincident with both the Spring
Bloom and initiation of near-surface warming from a wintertime minimum temperature of ca. 8
°C. Summertime values of $CH_2ClI$ and $CH_2I_2$ showed periodic variations similar to those observed
in Bedford Basin (section 3.2).
The lower temporal resolution of the study in Funka Bay (Shimizu et al., 2017), with sampling
only every 1 or 2 months, precluded detailed examination of timing. A gradual increase in $CH_3I$
appeared to start in March, during or towards the end of the Spring Bloom when surface water
temperatures were still close to their wintertime minimum of -1 to 2.5 °C. The seasonal increase
of $CH_2I_2$ and $CH_2ClI$ occurred later (May-June) at a time of rising water temperatures and low




nutrient levels with concentrations remaining elevated through the summer and decreasing to
wintertime levels in October.
The initial $CH_3I$ increase at a shallow station in the Kiel Fjord (Shi et al., 2014b) occurred in March,
and was closely linked in time to seasonal increases of solar radiation, temperature (winter
minimum of 0 °C) as well as Chl *a* and the springtime drawdown of nitrate. Lagged correlation
analysis showed similarly strong correlations of $CH_3I$ with both temperature and solar radiation.
However, the springtime increase of $CH_3I$, and its annual cycle, lagged temperature by ca. 1 month
leading *Shi et al.* (2014b) to attribute causality to solar radiation.
The observation of a rapid increase in the production rate of $I^-$ within phytoplankton cultures
(diatoms and prymnesiophytes) when they enter stationary and, especially, senescent phases
(Bluhm et al., 2011) is potentially relevant to the observed seasonality of iodocarbon formation.
The reduction of iodate to iodide was suggested to be due to release of precursors, such as reduced
sulphur species, to surrounding culture medium in association with a loss of membrane integrity
by stressed cells or as a result of viral lysis. *Hughes et al.* (2011) also reported studies with cultures
of *Prochlorococcus marinas* in which accumulation of $CH_3I$ commenced when cultures became
senescent. We note that significant iodocarbon accumulation in Bedford Basin was confined to
summertime when DIN was depleted (see fig. 2) and when cells may have been stressed or subject
to viral lysis, perhaps similar to later stages of batch culture experiments.
We therefore hypothesize that seasonal nitrate drawdown leads to increased supply of iodide to
surface waters which can, in turn, lead to increased formation of iodine atoms, HOI and $I_2$ as
precursors for iodocarbon formation by both photochemical and haloform reaction pathways
(Martino et al., 2009; Moore and Zafiriou, 1994). Whereas the supply of iodide may be one key
control, it is likely that variations in light intensity and water temperature also contribute to the





overall seasonality of the production rate of methyl iodide. For example, light can influence
formation of $CH_3I$ directly (e.g. Moore and Zafiriou, 1994; Richter and Wallace, 2004). Light can
also influence iodocarbon production indirectly, for example by producing oxidants such as $H_2O_2$
to promote oxidation of iodide by haloperoxidases (Hill and Manley, 2009) or by altering the
quality of dissolved organic matter. The time-series of $CH_2I_2$ and $CH_2ClI$ from very shallow (< 4
m), nearshore waters of the Kattegat, Sweden (Klick, 1992) and the Baltic Sea, Germany
(Orlikowska and Schulz-Bull, 2009) showed peaks in April/ May and again in September/October,
with low concentrations throughout summer. This contrasts with the deeper water columns of
Bedford Basin, Funka Bay and the English Channel where concentrations remain elevated
throughout summer. This likely reflects dominance of photolytic loss over production within very
shallow water columns exposed to summertime light intensities and long periods of daylight. Sub-
surface production coupled with vertical mixing may explain the summertime persistence in
deeper water columns.
4.3.2 Sub-seasonal periodicity
The key feature at the sub-seasonal scale during summer is periodicity of near-surface iodocarbon
concentrations which was observed in both the English Channel and Bedford Basin. There was no
obvious pattern linking maxima and minima of the three compounds. For example, whereas in
Bedford Basin, peaks in all three compounds tend to coincide, their relative amplitudes are variable.
In the English Channel, peaks of $CH_2I_2$ and $CH_2ClI$ also appeared to coincide, however $CH_3I$
variability was largely decoupled (Archer et al., 2007). The periodicity of iodocarbon
concentrations may be a consequence of the internal coupling of iodine cycling reactions, together
with depth- and light-dependent photolytic degradation, as much as the result of external forcing.
4.3.3 Interannual variability

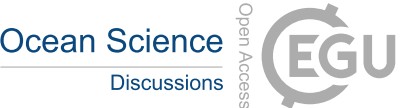

The Bedford Basin time-series is unique in having high temporal resolution sampling (weekly)
over three annual cycles which allows interannual variability to be examined for the first time. The
most obvious interannual difference was in the behavior of methyl iodide. In particular, 2016 was
markedly different in that only a single peak was observed in late August, whereas the summers
of 2015 and 2017 were marked by 3-4 quasi-periodic, multi-week maxima. As noted already, the
English Channel time-series of $CH_3I$ did not exhibit this behavior. The shallow-water time-series
of $CH_3I$ in the Kiel Fjord and coastal Baltic Sea (Orlikowska and Schulz-Bull, 2009; Shi et al.,
2014b) also did not exhibit this type of variability. Because the cause of the periodicity itself is not
understood or explained, discussion of reasons for its interannual variation must be highly
speculative. One clear difference of 2016 relative to the other two years, was the lower
summertime precipitation and associated lack of near-surface salinity stratification. The temporal
behavior of $CH_3I$ in 2016 might therefore be related to altered near-surface mixing dynamics
within Bedford Basin, or alternatively, to decreased delivery of key precursors (e.g. DOM) from
land via rivers and wastewater.
4.4 Vertical distributions and subsurface temporal variability
Figure 4 shows the near-surface concentration variations of the VOIs. For $CH_3I$, concentrations
were almost always uniform between 1, 5 and 10 m. For $CH_2ClI$, the concentrations at 1 and 5m
were usually very similar (average difference -4.1 %; median -2.5 %), however concentrations at
10m depth were noticeably lower for periods of time. For $CH_2I_2$, the highest concentrations were
observed at a depth of either 5 or 10 m, with concentrations at 5 m occasionally peaking at very
high levels (e.g. 250-350 pmol $L^{-1}$) for short periods (less than one week). Concentrations at 1 m
were almost always lower than at 5 m, with the percentage reduction relative to 5 m averaging 52 %





in summer. Concentrations at 10 m, on the other hand, were generally the same or higher as those
measured at 5 m (with the exception of the previously mentioned, short-lived peaks).
These results are consistent with earlier studies of vertical profiles in the open ocean (e.g. (Moore
and Tokarczyk, 1993; Yamamoto et al., 2001)) as well as with model predictions (Jones et al.,
2010; Martino et al., 2006). In particular, our results are consistent with the quantitative predictions
of a mixed-layer model (Jones et al., 2010) that $CH_2ClI$ would typically be near-uniform within
the upper 6 m of the water column, whereas photolytic decay could remove up to 100 % of the
$CH_2I_2$ over that depth range, depending on time of day and conditions.
4.5 Temporal variability in near-bottom water (60 m)
Temporal variations of VOIs in near-bottom waters (60 m), were of lower amplitude except for
the already noted variation of $CH_3I$ during the winter of 2015-2016 (fig. 4d). From June to
December 2015, $[CH_3I]$ increased steadily (concentration change, $\Delta C = 20$ pmol $L^{-1}$) so that
concentrations exceeded those in surface waters from October 2015 until end of March 2016. This
did not occur in the two subsequent years, with $[CH_3I]$ remaining constant through the remainder
of 2016 and showing a smaller seasonal increase ($\Delta C = 8$ pmol $L^{-1}$), confined to early summer, in

17    2017.

Concentrations of $CH_2ClI$ remained almost constant at <5 pmol $L^{-1}$ throughout the time-series with
the notable exception of abrupt (<1 week) increases in May and November 2016 and December
2017. These increases ($\Delta C = 2$-$5$ pmol $L^{-1}$) corresponded with increases of salinity (and $O_2$)
indicating that they were linked to intrusions of saltier, near-surface waters from offshore. The
subsequent concentration declines (with estimated half-life, assuming 1st order kinetics, of order
70 days) reflects loss due to mixing or, more likely, by reaction or microbial degradation within



the water column and/or sediments. The same three intrusions drove abrupt increases of $CH_2I_2$
with amplitude ca. $1.5 - 2$ times higher than those for $CH_2ClI$, consistent with near-surface
concentration ratios (see section 3.2). However, $CH_2I_2$ showed additional, higher amplitude
variability, unrelated to the bottom water intrusions, which is discussed in detail below.
In 2015, a gradual but high-amplitude increase of $CH_3I$ ($\Delta C = 20$ pmol $L^{-1}$) from June through
October, paralleled a steady decline in oxygen, suggesting $CH_3I$ production linked to degradation
of organic matter (apparent production rate of 0.06 pmol $L^{-1}$ day$^{-1}$ from June to December). This
is ca. 20 times lower compared with the annual averaged $P_{net}$ of $CH_3I$ (see section 3.4) in surface
water of ca. 1 pmol $L^{-1}$ day$^{-1}$. This initial part of the time-series is consistent with results from
short-term incubation experiments (3 days duration) conducted by *Hughes et al.* (2008) with
biogenic marine aggregates. They observed increasing concentrations of mono-iodinated
iodocarbons, including $CH_3I$, but no corresponding increase in the dihalogenated compounds such
as $CH_2I_2$ and $CH_2ClI$. The results suggested alkylation of inorganic iodine or breakdown of higher
molecular mass organohalogens as production pathways and, following *Amachi et al.* (2001), it
was suggested that microbial degradation increases the supply of precursors.
However, as $O_2$ concentrations declined further from October through late December, the
concentration of $CH_3I$ stabilized and the $CH_2I_2$ concentration increased markedly from ca. 2 to 12
pmol $L^{-1}$. From January through April 2016, $CH_3I$ levels decreased rapidly, in concert with
increasing $O_2$ concentrations and decreasing salinity (see fig. 4) as a result of progressive vertical
mixing with overlying waters with lower $CH_3I$ concentrations. Over this period, the $CH_2I_2$
concentrations at 60 m remained almost constant (due to the smaller vertical gradient of
concentration), before decreasing again in summer, in parallel with the seasonal decline of $O_2$.





The combined time-series is suggestive of a switch of production mechanism from an alkylation
pathway producing mono-iodinated compounds ($CH_3I$) to a haloform-type reaction which
produces $CH_2I_2$. The apparent "switch" took place in October, when oxygen concentrations
dropped below ca. 90 µmol $kg^{-1}$, although whether the switch was related to redox conditions in
the water column or sediments, the speciation and availability of iodine, or the availability of
suitable organic precursors and/or enzymes cannot be determined.
There was no significant near-bottom iodocarbon production throughout 2016, especially for $CH_3I$,
and therefore no "switch", for reasons that are not clear given that $O_2$ concentrations declined
through most of the summer and fall, until interrupted by an intrusion in November. It can only be
speculated that the lack of production might be linked to the limited and time-restricted period of
production of $CH_3I$ in near-surface waters (fig. 4a) in 2016. In 2017, there was moderate sub-
surface production of $CH_3I$, associated with $O_2$ consumption, and again an apparent "switch" to
$CH_2I_2$ production as marked by a plateau of $CH_3I$ concentrations at the same time as $CH_2I_2$
concentrations started to increase. The switch took place earlier in the year and at significantly
higher $O_2$ concentrations (175 µmol $kg^{-1}$) than in 2015. Closer examination suggests, however,
that the $CH_3I$ and $CH_2I_2$ time-series in 2017 may have been impacted more by intrusions and
mixing of water with different origins than by changing reaction mechanisms, as explained below.
Figure 4 shows that the moderate increase of $CH_3I$ at 60m, observed from April through early July
2017, was followed by a concentration plateau lasting ca. 6 weeks and a subsequent decrease to
background levels over a period of <4 weeks. Close inspection of figure 4 suggests that the plateau
of $CH_3I$ was linked with a mid-depth intrusion of salty, offshore waters (as denoted by the 31
salinity contour). The period from the beginning of July to mid-August 2017 was marked by a
pronounced increase in the rate of warming at 60 m, a switch from declining to increasing salinity,





and a reduction in the rate of oxygen concentration decline. The mid-depth salinity maximum must
have been due to an intrusion of saltier water from offshore. Generally, intruding waters are dense
enough to sink to the bottom of Bedford Basin so that they mix with and displace the low-oxygen
waters that develop there, as seen at other times in the time-series. However, the altered trends in
T, S and $O_2$ at 60m over the 6-week period in 2017 are consistent with mixing between a newly-
introduced offshore end-member at mid-depth and the pre-existing deeper water of Bedford Basin.
It is therefore likely that mixing also contributed to the sudden increase of $CH_2I_2$ and the "plateau"
of $CH_3I$ concentrations, which both occurred at exactly the same time. A very small but significant
increase in $CH_2ClI$ ($\Delta C$=1-2 pmol $L^{-1}$) commenced at the same time. At the end of this period, in
mid-August, the rate of warming and salinity increase at 60 m decreased again, and the rate of
oxygen decline increased, suggesting that the intrusion's impact at 60 m had lessened. At this time,
$CH_2I_2$ and $CH_3I$ concentrations decreased immediately, returning to background levels within
about 1 month (1st order half-life of 65 days for $CH_2I_2$ and 14 days for $CH_3I$). A corresponding
decrease in [$CH_2ClI$] concentration appeared to start 4-6 weeks later with the rate of decline
matching that observed after the three intrusions discussed earlier (1st order half-life of 70 days).
This detailed discussion of the observed temporal variability emphasizes that a variety of
underlying physical and biogeochemical mechanisms are responsible for the observed seasonal,
sub-seasonal and interannual variability of the three iodocarbons. The high amplitude signals
observed in Bedford Basin suggest that these complex and variable processes could be resolved,
in detail, with more targeted, higher-resolution sampling, and would be amenable to examination
with models, so that the variability itself would be useful for model validation. Separation of the
multiple contributing factors and processes that underlay observed temporal variability will,
however, require a seasonally-resolved time-series of experimental studies that are linked to a





physical model that can represent both mixing dynamics and key aspects of aquatic iodine
chemistry.
4.6 Sea-to-air fluxes
The temporal variation of the sea-to-air flux of $I_{org}$ and its relative contribution from the three
iodocarbons is shown in figure 5 and table 4.
Consistent with earlier time-series (excluding those from very shallow waters, see section 4.2), the
sea-to-air flux of iodocarbons is generally highest in summer/fall. However, high wintertime fluxes
are also possible, as shown in 2017 when there was a large efflux of $CH_2I_2$ (averaging 41 nmol m$^{-}$
$^2$ d$^{-1}$; table 4), due to both strong winds and relatively high concentrations. The fluxes of $CH_3I$ and
of $CH_2ClI$, on the other hand, were always higher in summer (ca. 3-5X and 10X higher,
respectively).  Similar results were presented by *Shimizu et. al* (2017) with the total iodine flux in
Funka Bay in summer being > 5 times that in winter.
Our estimated fluxes of $CH_3I$ (8.7nmol m$^{-2}$ d$^{-1}$, table 5) are in the range of emissions calculated for
coastal and continental shelf water in similar latitudes (9.6  and 11.9 nmol m$^{-2}$ d$^{-1}$; Archer et al.,
2007; Shimizu et al., 2017 respectively). The average flux of $CH_3I$ reported by Jones et. al (2010),
from the west of Ireland, was 4 times higher but based on a sampling period of only 1 month during
summer only. Sea-to-air fluxes of $CH_2ClI$ from the same studies were very similar to the fluxes
from Bedford Basin that we calculated. For $CH_2I_2$, the annual averaged flux (17.8 nmol m$^{-2}$ d$^{-1}$) in
Bedford Basin is 5 times higher than in west English Channel, and similar as in Funka Bay, Japan.
The total, annual $I_{org}$ sea-to-air flux from Bedford Basin averaged 62 nmol m$^{-2}$ d$^{-1}$, approximately
8 times higher than the flux of $CH_3I$. This flux is ca. 40% higher than the annual fluxes reported
for the Western English Channel and Funka Bay, Japan (Archer et al., 2007; Shimizu et al., 2017)



and the relative contribution of polyiodinated compounds to the total flux was also slightly higher
in Bedford Basin (table 5).
As shown in figure 5 and table 4, however, the total $I_{org}$ flux is subject to significant interannual
variability, which could not be assessed in earlier studies. Notably, the $I_{org}$ flux in 2016 was 2-3X
lower than in 2015 and 2017. A comparison of wind-speeds and concentrations showed that
although both factors contributed to interannual flux variation, the effect of wind speed dominated,
with winds during summer/fall of 2016 being 1-2 m s$^{-1}$ lower than average.
4.7 Production rate of $CH_3I$
The annual mean production rate of $CH_3I$ in this study, estimated using equation 1, was 1.0 pmol
L$^{-1}$ day$^{-1}$ (ranging from -1.6 to 8.5 pmol L$^{-1}$ day$^{-1}$, see section 3.4 and fig. 6). This is comparable
with the global average production rate estimated by *Stemmler et al.* (2013) (1.64 pmol L$^{-1}$ day$^{-1}$),
for which 70% was produced via a photochemical mechanism. Based on data presented by *Archer*
*et al.* (2007), the annual mean production rate of $CH_3I$ in the western English Channel was ca. 2
pmol L$^{-1}$ day$^{-1}$ (range: -0.2 to 6 pmol L$^{-1}$ day$^{-1}$). Here it should be noted, that their "minimum gross
production rate" is equivalent to $P_{net}$ in this study and in *Shi et al.* (2014a). In contrast, *Shi et al.*
(2014b) estimated a considerably lower annual mean net production rate in the Kiel Fjord of ca.
0.1 pmol L$^{-1}$ day$^{-1}$ (maximum of 0.8 pmol L$^{-1}$ day$^{-1}$). The maximum production rates from the Kiel
Fjord study were based on monthly average concentrations, and therefore expected to be smaller.
However, *Shi et al.* (2014a) also conducted weekly incubation experiments in Kiel Fjord which
gave *in vitro* values of $P_{net}$ which were closely comparable with the field-based estimates.

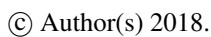


The lower values of $P_{net}$ in the Kiel Fjord compared with both Bedford Basin and the English
Channel must reflect either differences in gross production (e.g. due to differences in the supply
of precursors and reactants such as iodide) or differences in other, uncharacterized losses. An
additional, poorly characterized loss process, possibly microbial degradation, was in fact observed
in the Kiel Fjord incubation experiments (Shi et al., 2014a). On the other hand, incubation
experiments conducted with additions of labelled methyl iodide ($^{13}CD_3I$) to Bedford Basin surface
waters (data not shown) showed no such losses. We therefore hypothesize that the lower $P_{net}$ in
Kiel Fjord is a result of higher microbial degradation of $CH_3I$ in that very shallow (12 m deep),
nearshore environment.
5.  Conclusions, Implications and Further Work
The key findings of the time-series sampling of Bedford Basin are as follows:
Iodocarbon concentrations in near-surface waters showed strong seasonal variability and
similarities and differences in their correlation with temporal variations of potentially-related
properties and causal factors in comparison to other coastal time-series. Based on the time-series
and lab studies, we hypothesize that the production of iodocarbons is linked to accelerated
reduction of iodate to iodide under conditions of post-bloom phytoplankton senescence, nutrient
stress and/or viral lysis following the seasonal disappearance of nutrients.
The observed vertical variation of $CH_2I_2$ and $CH_2ClI$ within the upper 10m is consistent with the
more rapid photolysis of $CH_2I_2$ and is affected by near-surface stratification and mixing associated
with variable freshwater input.
Seasonality of iodocarbon concentrations in Bedford Basin is similar to that observed in coastal
time-series from the English Channel and Funka Bay, Japan, but does not exhibit the mid-summer



minimum in the concentration of polyiodinated compounds observed in very shallow time-series
(<10m), which likely reflect the dominance of photolytic decay in such shallow water columns.
Quasi-periodic variability of all three compound with a characteristic timescale of 4-5 weeks was
observed in summertime near-surface waters and is similar to variability observed in the western
English Channel.
Near-surface interannual variability was particularly pronounced for $CH_3I$, with only one, short-
lived near-surface concentration maximum occurring in 2016. The reasons for the variability are
unclear but may be a consequence of anomalously low rainfall and an associated reduction in the
supply of terrestrial organic matter during that summer.
A lack of spatial variation of near-surface iodocarbon concentrations within Bedford Basin is
consistent with earlier studies and confirms that air-sea fluxes are not influenced directly by
macroalgal sources, even in nearshore waters.
Average annual sea-to-air fluxes (62 nmol m$^{-2}$ d$^{-1}$) of total volatile organic iodine ($I_{org}$) were
comparable to or slightly higher than those oberved in Funka Bay, Japan, and the English Channel.
The polyiodinated compounds contributed ca. 85 % of the total flux which was also higher than in
the other two time-series and confirms that the sea-to-air flux of polyiodinated compounds such as
$CH_2I_2$ and $CH_2ClI$ are larger than those of $CH_3I$ in coastal waters. The fluxes were subject to strong
interannual variability (2-3X) as a result, mainly, of wind-speed variability but also variability of
surface concentrations.
Concentrations and temporal variability were smaller in near-bottom waters (60 m). However, the
60m time-series showed evidence for $CH_3I$ production associated with the decay of sinking organic
matter. There was a suggestion in the time-series that iodocarbon production could "switch" from
$CH_3I$ production (e.g. by alkylation of organic matter) to production of $CH_2I_2$ (e.g. by a haloform



type reaction), after a period of ca. 5 weeks. Other increases in $CH_2I_2$ and $CH_2ClI$ at 60m were
associated with intrusions of saltier, near-surface waters carrying higher iodocarbon concentrations.
Concentrations decreased following these intrusions with half-lives, assuming 1st order kinetics,
of 14 days ($CH_3I$), 65 days ($CH_2I_2$) and 70 days ($CH_2ClI$). These half-lives are likely lower bounds
for chemical/ microbial degradation of these compounds given that concentrations might also be
reduced as a result of vertical mixing.
Net production rates of $CH_3I$ averaged 1.0 pmol $L^{-1}$ $day^{-1}$ and were similar to rates estimated in
the English Channel but an order of magnitude higher than rates estimated and measured in shallow,
nearshore waters of the Kiel Fjord. The low net production in the Kiel Fjord is hypothesized to be
the result of high microbial degradation rates, as suggested by *in vitro* incubations. These were not
observed in Bedford Basin,
In general, the time-series confirms that there are at least two pathways of $CH_3I$ production
(photochemical and production via an alkylation pathway) operating in coastal waters, and the
photochemical production rate in near surface water is more than one order of magnitude higher
than the alkylation pathway rate in near-bottom water. Coastal waters can also support high rates
of production of polyiodinated compounds. The seasonality and variability of production of both
mono- and poly-iodinated compounds may be a function of the rate of supply of iodide. The higher
rates of $CH_2I_2$ and $CH_2ClI$ production in comparison with offshore waters may, additionally,
reflect a greater supply of organic precursor compounds (e.g. from land). Alternatively, it may
reflect differences in inorganic iodine cycling between coastal and offshore waters.
The very high amplitude concentration variations encountered in Bedford Basin, coupled with its
relative accessibility for high-frequency sampling and constrained, yet variable, physical
exchanges make Bedford Basin a useful location to investigate complex iodine cycling dynamics.



To-date, the multiple production pathways and relative complexity of iodine biogeochemistry have hindered progress towards understanding the underlying dynamics that result in varying spatial and temporal fluxes of iodine between the ocean and atmosphere.

Based on these results, we suggest that progress in detailed understanding can now be made through the implementation of a more comprehensive sampling strategy (higher vertical resolution and inclusion of inorganic iodine speciation measurement), coupled to a biogeochemical model of Bedford Basin that includes iodine chemistry (e.g. Stemmler et al, 2013) and a time-series of targeted experimental studies conducted in the context of the time-series (see Shi et al., 2014b). In other words, Bedford Basin may be an ideal location and time-series upon which to base a multi-investigator campaign focused on understanding environmental controls on volatile iodine cycling and improving their representation in models.

Acknowledgments

This work was funded by the Canada Excellence Research Chair in Ocean Science and Technology at Dalhousie University. Sampling of a Bedford Basin time-series was supported by MEOPAR Observation Core and Bedford Institute of Oceanography. The authors thanks to Richard Davis, Anna Haverstock and the crew of the Sigma-T. Assistance and guidance in the laboratory from Claire Normandeau is  also acknowledged. We also thank Environment and Climate Change Canada for windspeed data and precipitation data, and NASA for modeled solar irradiance data.

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





1   **Table 1.** correlations between individual iodocarbons based on weekly data and monthly average.

|  | weekly | | | monthly | | |
|---|---|---|---|---|---|---|
|  | $CH_3I$ | $CH_2ClI$ | $CH_2I_2$ | $CH_3I$ | $CH_2ClI$ | $CH_2I_2$ |
| $CH_3I$ (1m) | 1.0 | | | 1.0 | | |
| $CH_2ClI$ (1m) | 0.4 | 1.0 | | 0.7 | 1.0 | |
| $CH_2I_2$ (1m) | 0.0 | 0.3 | 1.0 | 0.1 | 0.4 | 1.0 |
| $CH_3I$ (5m) | 1.0 | | | 1.0 | | |
| $CH_2ClI$ (5m) | 0.4 | 1.0 | | 0.5 | 1.0 | |
| $CH_2I_2$ (5m) | 0.1 | 0.4 | 1.0 | 0.1 | 0.6 | 1.0 |
| $CH_3I$ (10m) | 1.0 | | | 1.0 | | |
| $CH_2ClI$ (10m) | 0.7 | 1.0 | | 0.7 | 1.0 | |
| $CH_2I_2$ (10m) | 0.1 | 0.3 | 1.0 | 0.2 | 0.4 | 1.0 |



1    **Table 2.** correlations between iodocarbons and potentially relevant parameters.

|  | weekly | | | monthly | | |
|---|---|---|---|---|---|---|
| 1m | $CH_3I$ | $CH_2ClI$ | $CH_2I_2$ | $CH_3I$ | $CH_2ClI$ | $CH_2I_2$ |
| SST | 0.5 | 0.4 | 0.0 | 0.7 | 0.6 | 0.1 |
| SSS | 0.1 | 0.0 | 0.0 | 0.1 | 0.1 | 0.0 |
| Oxygen | 0.3 | 0.2 | 0.1 | 0.3 | 0.2 | 0.1 |
| Flu | 0.0 | 0.0 | 0.0 | 0.1 | 0.0 | 0.0 |
| PAR | 0.1 | 0.1 | 0.0 | 0.1 | 0.3 | 0.0 |



1    **Table 3.** Concentration (pmol L$^{-1}$) of iodocarbons in various measured at near shore locations of

2    Bedford Basin in 2017 (see Fig. 1a).

|  | $CH_3I$ | $CH_2ClI$ | $CH_2I_2$ |
|---|---|---|---|
| [1]Tufts cove | 5.8 | 35.5 | 8.8 |
| [2]Wrights cove | 6.7 | 20.6 | 12.5 |
| [3]Sackville | 3.8 | 6.5 | 4.9 |
| [4]Mill cove | 8.3 | 28.0 | 18.6 |
| [5]Fairview cove | 6.2 | 26.3 | 6.4 |
| Middle of Bedford Basin | 6.1 | 37.6 | 6.3 |



1  **Table 4.** Seasonal variation of total sea-to-air fluxes of iodocarbons (nmol m$^{-2}$ d$^{-1}$). In spring 2015,

2  we have substituted the flux of iodocarbons based on the average values from spring 2016 and

3  spring 2017. Higher flux values of iodocarbons in each year are marked in red.

| year | season | CH$_3$I | CH$_2$ClI | CH$_2$I$_2$ | Total iodine |
|------|--------|------|--------|--------|--------------|
| 2015 | *Spring | 3.0 | 5.0 | 8.1 | 24.2 |
|      | Summer | 17.6 | 42.0 | 25.7 | 110.8 |
|      | Fall | 13.7 | 38.1 | 28.6 | 109.1 |
|      | Winter | 2.8 | 3.6 | 13.8 | 34.0 |
|      | Annual | 13.8 | 34.8 | 25.0 | 98.6 |
| 2016 | Spring | 3.0 | 7.6 | 15.2 | 40.9 |
|      | Summer | 9.9 | 27.7 | 11.2 | 60.0 |
|      | Fall | 5.6 | 6.9 | 2.1 | 16.8 |
|      | Winter | 3.2 | 2.6 | 8.7 | 23.2 |
|      | Annual | 5.7 | 12.2 | 9.1 | 36.2 |
| 2017 | Spring | 3.1 | 3.0 | 2.5 | 11.1 |
|      | Summer | 12.2 | 26.3 | 21.0 | 80.4 |
|      | Fall | 13.9 | 27.3 | 17.3 | 75.8 |
|      | winter | 5.3 | 15.4 | 40.7 | 102.0 |
|      | Annual | 8.7 | 17.8 | 19.2 | 64.9 |



**Table 5.** Comparison of sea-to-air flux (nmol m$^{-2}$ d$^{-1}$) of total organic iodine in different studies.
Data from Archer et al. (2007) is 1 year annual average, from Shimizu et al. (2017) is 3 years
annual average, from Shi et al. (2014b) is 2 years annual average (only the flux of CH$_3$I) and from
our study is ca. 3 years average.

| | English Channel | | Funka Bay | | Kiel Fjord | | Bedford Basin | |
| | Archer et al.(2007) | | Shimizu et al. (2017) | | Shi et al. (2014) | | This Study | |
| Season | Total | %-CH$_3$I | Total | %-CH$_3$I | Total | %-CH$_3$I | Total | %-CH$_3$I |
|---|---|---|---|---|---|---|---|---|
| Spring | | | 12.3 | 46.7 | 2.8 | | 23.6 | 12.7 |
| Summer | | | 79.3 | 14.0 | 5.2 | | 82.4 | 15.8 |
| Fall | | | 51.7 | 36.0 | 2.2 | | 67.2 | 16.5 |
| Winter | | | 18.0 | 27.0 | 0.2 | | 60.7 | 6.8 |
| Annual | 42.6 | 27.9 | 43 | 22.3 | 3.3 | | 61.7 | 14.1 |


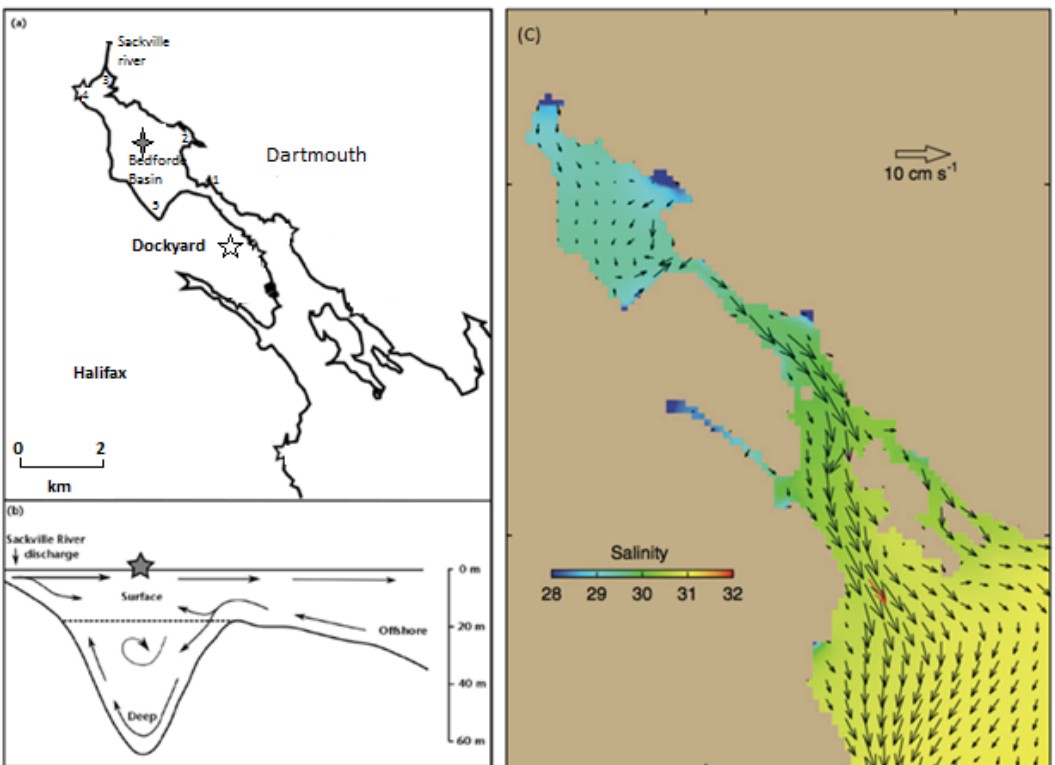

**Figure 1.** (a): Main sampling location (grey star) and near-shore sampling locations 1-5 (1: Tufts
cove; 2: Wrights cove; 3: Sackville; 4: Mill cove; 5: Fairview cove) in Bedford Basin; (b): two
layered flow in Halifax Harbour, adapted from *Kerrigan et al.* (2017); (c): horizontal circulation
of water in Halifax Harbour from *Shan et al.* (2011), using annual mean currents and velocities.





**Figure 2.** Seasonal variation of (a) chlorophyll *a* (blue line for 1 meter and red line for 5 meter);
(b) DIN; (c) near surface salinity, (d) precipitation and (e) windspeed in Bedford Basin from
January 2015 to Dec 2017.





**Figure 3.** Seasonal patterns of environmental and biological variables in Bedford Basin from
January 2015 to Dec 2017. (a) temperature; (b) salinity (showing contour lines for 30.5 and 31);
(c) dissolved oxygen; (d) chlorophyll fluorescence.



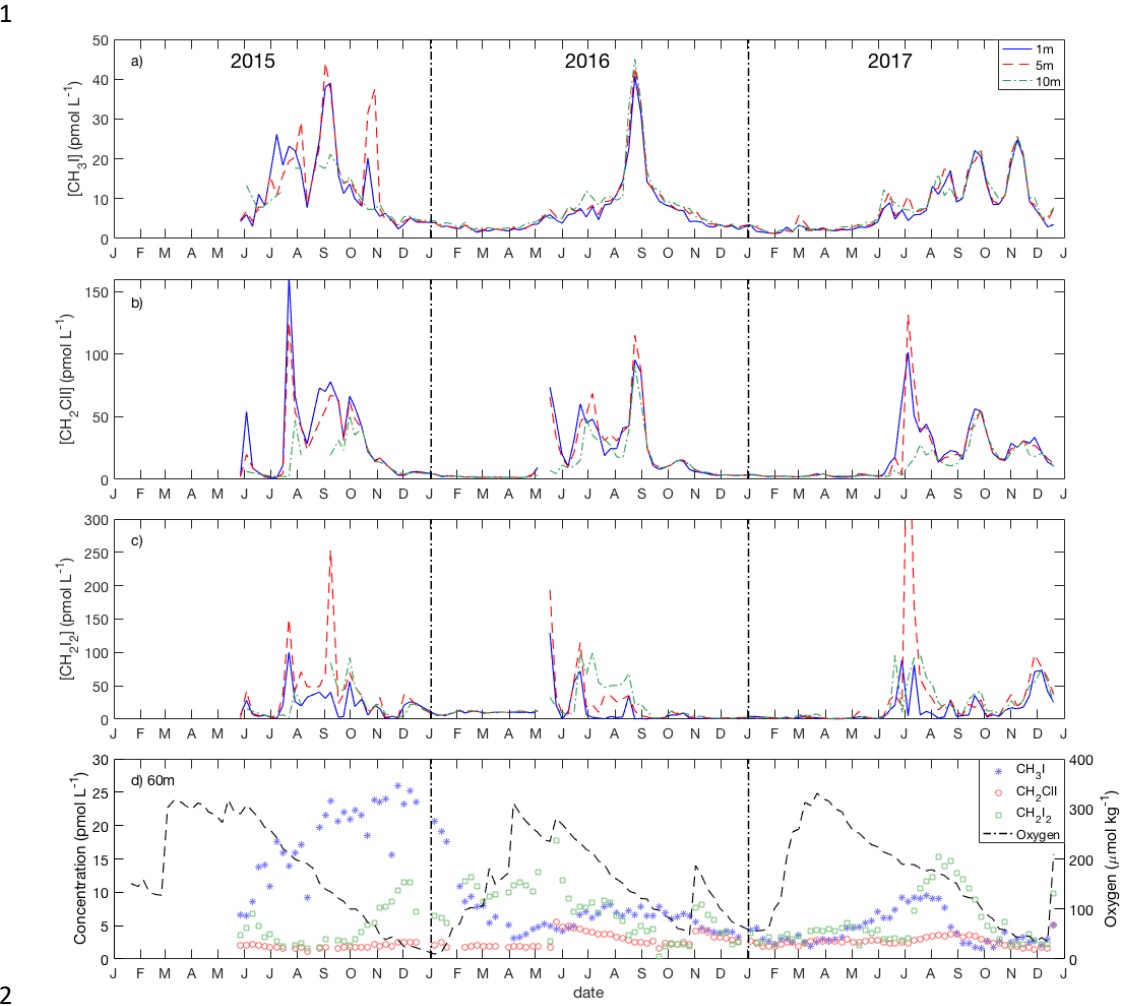

**Figure 4.** Seasonal variation of Iodocarbons in the Bedford Basin at 1 meter (blue line), 5 meter (red line) and 10 meter (green line) from May 2015 to December 2017: (a) $CH_3I$; (b) $CH_2ClI$ and (c) $CH_2I_2$. (d): time-series of near-bottom water (60m) of iodocarbons and dissolved oxygen.



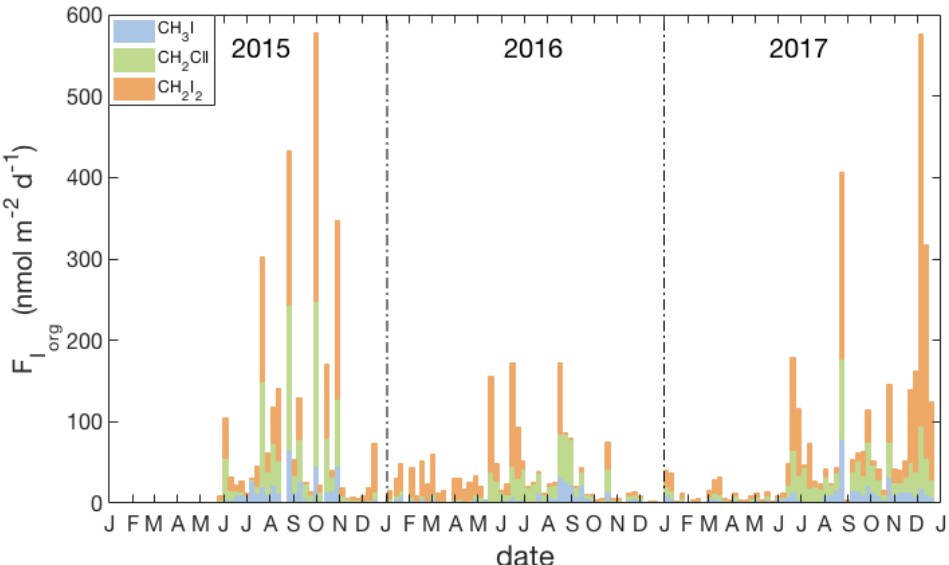

3  **Figure 5.** Sea-to-air flux of $I_{org}$: including relative contributions of individual compounds and
4  using the parameterization for transfer velocity of Nightingale et al. (2000).





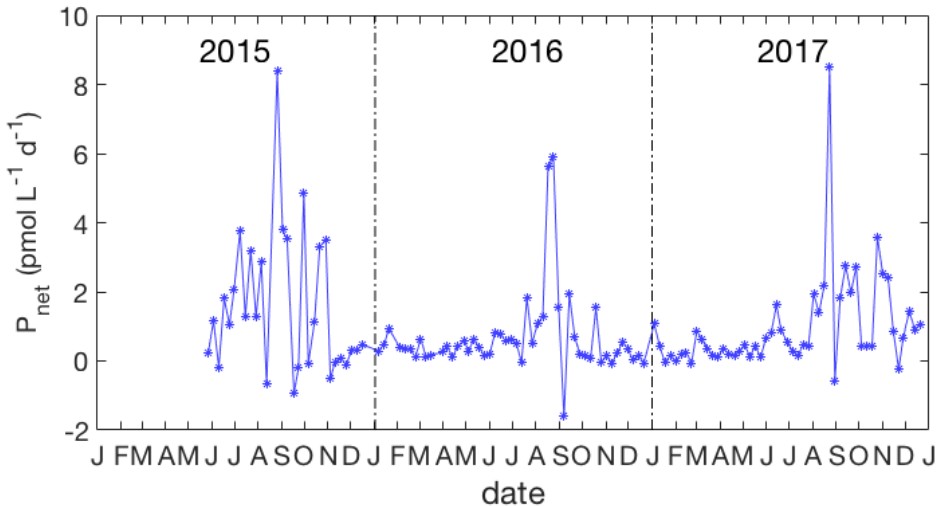

**Figure 6.** Variation of net production rate of CH$_3$I from 2015 to 2017.




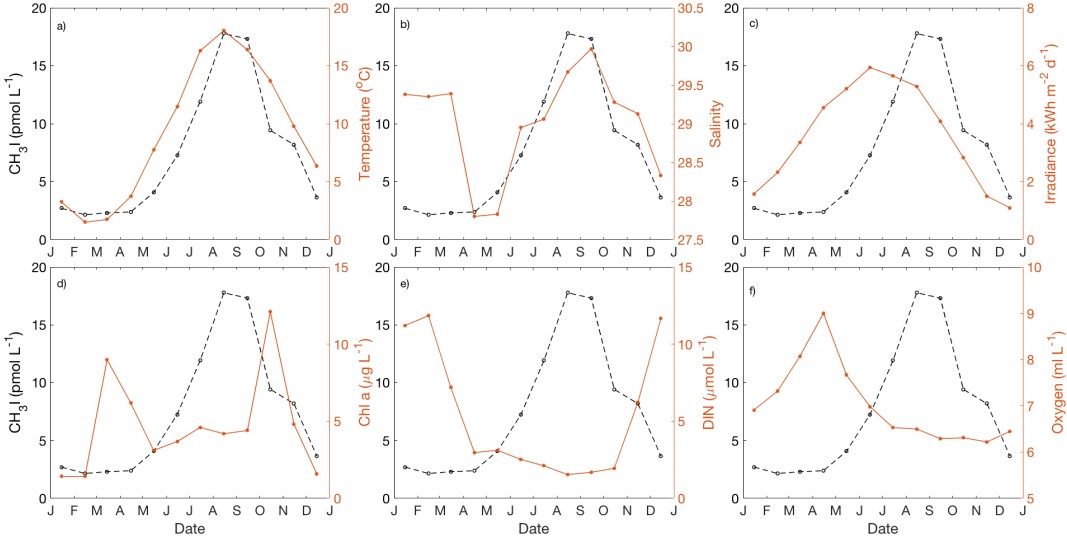

**Figure 7.** Annual cycle of (a) temperature, (b) salinity , (c) irradiance, (d) Chl *a*, (e) dissolved inorganic nitrogen (DIN) and (f) dissolved oxygen for near-surface water (1-5 m) in Bedford Basin. The black dashed line depicts the annual cycle of CH3I. The figures present the monthly mean values based on the data collected from May 2015 to December 2017.



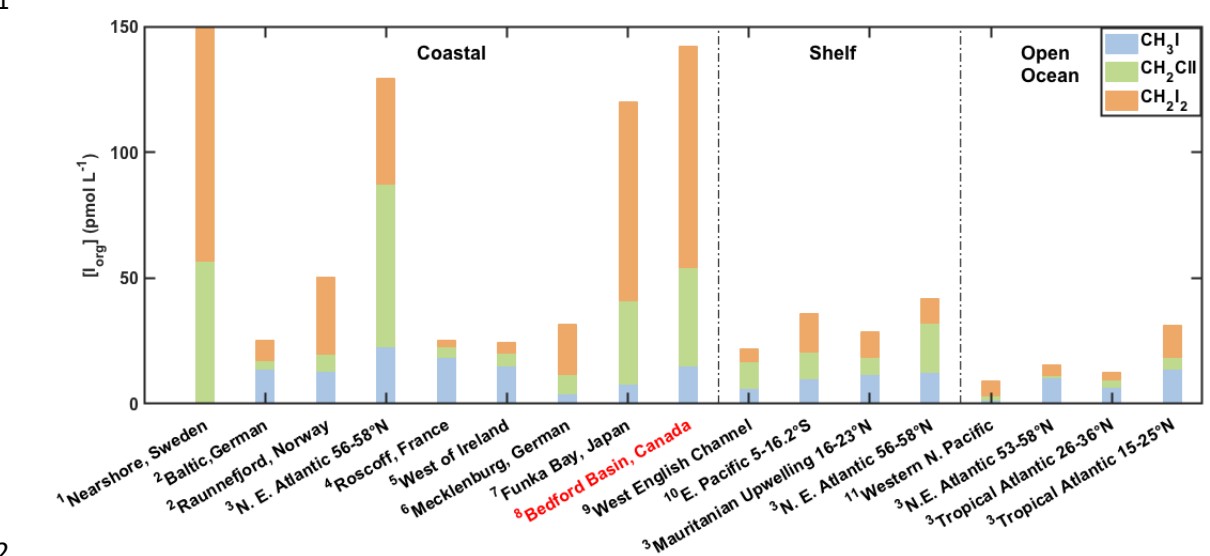

**Figure 8.** Contribution of Iodocarbons to total organic iodine (Iorg) in surface seawater from
different regions and studies:
1 : Klick, 1992; 2: Orlikowski et al., 2015; 3: Jones et al., 2010;4: Jones et al. 2009; 5: Carpenter
et al., 2000; 6: Orlikowski & Schulz-Bull, 2009; 7: Shimizu et al.,2017; 8: this study; 9: Archer et
al., 2007; 10: Hepach et al., 2016; 11: Kurihara et al., 2010.
