# Peer review of "A three year time-series of volatile organic iodocarbons in Bedford Basin, Nova Scotia"

_Ocean Science, 2018_

## Referee Comment (RC1) · Anonymous Referee #1 · 1 Aug 2018

General comment The manuscript "A three year time-series of volatile organic iodocarbons in Bedford Basin, Nova Scotia : a Northwestern Atlantic fjord, os-2018-70" by Shi and Wallace, measured the concentrations of $CH_3I$, $CH_2ClI$, and $CH_2I_2$ in seawater as well as bio-chemical compounds, such as oxygen and nitrogenous nutrient, at a fjord in NW Atlantic. They found that dihalomethanes are the main constituent among the three iodocarbons. They estimated a sea-to-air flux of $CH_3I$, and net production rate of $CH_3I$ in the surface layer. In the deep layer at 60 m depth, the $CH_3I$ concentration gradually increased and plateaued for several weeks, and then, the $CH_2I_2$ concentration increased in duration of the $CH_3I$ plateau. They suggested a "switch" of iodocarbon production type from "methylation type", which produces $CH_3I$ only, to "haloform type",

which mainly produces CH2I2 with decreasing oxygen concentration. This is the first report of weekly measurement of iodocarbon in seawater from the surface to bottom layers at an observation station for three years. The long-term observation data is valuable for publication, however, the observed trends were similar to the previous observations. Note that the increase of similar knowledge about the temporal trends of iodocarbon in coastal area is valuable. The interpretation of production of iodocarbon via HOI or I2 formation by this study is also similar to the previous studies. This study supported the previous interpretations. The original finding of this study is a "switch" of iodocarbon production type from "methylation type" to "haloform type", based on the concentration changes in iodocarbon at 60 m depth. And, the authors explained that several changes in iodocarbon concentration were attributed to water intrusions from open ocean to the fjord.

Important point (1) Nevertheless of importance of concentration changes at 60 m depth, it is difficult to follow the explanation described in the manuscript comparing with the Figure 4d (iodocarbon at 60 m depth) and Figure 3 (salinity profile). It is necessary to add some detailed figures to explain timings of concentration changes of iodocarbon and water intrusions (based on analysis of salinity changes), etc.

Important point (2) Authors calculated the mass balance of CH3I in the top 10m of water column except for winter, however, mass balance must be calculated in the surface mixed layer defined by vertical variance of density. The thickness of the surface layer, in which water can contact with air, is crucial to estimate the balance between sea-to-air out-flux and net-production in the surface layer. The weekly observation of this study make it possible to set the thickness appropriately. I believe that it makes this paper more valuable.

Specific comments Abstract and discussion "hypothesis that near-surface iodocarbon production is linked to reduction of iodate to iodide" I agree that dihalomethane production is supposed to associated with I2 (or HOI) production and subsequent reaction with organic matter. However, there is no evidence that I2 production is attributed to

iodate reduction. Some previous studies supposed that I2 is produced as a result of iodide oxidation. Iodate reduction might occur in the surface seawater with depleting DIN. I can approve the hypothesis that near-surface iodocarbon production is linked to reduction of iodate to iodide at a discussion chapter, however, the hypothesis should not be described in the abstract.

Abstract Line 24: (2- 3 X) Does it mean "two to three times" ? I am not sure that 2- 3 X is appropriate in scientific writing.

Introduction Line6: "CH2ClI (hours) and CH2I2 (minutes)" is correct.

Page 7, Line 8: dissolved inorganic nitrogen (DIN) The definition of DIN ( = NO3 + NO2 + NH4) should be described at here, instead of page8 line 22.

Page 8, line 6- 12, etc. The authors explained degradation of organic matter using oxygen concentration. I think that AOU is better to explain the degradation of organic matter.

Page 9, Line 22: "Missing from the bottom water time-series, were the ca. 1 month duration variations seen in summertime surface water." I cannot understand the meaning of this description.

Page 16, line 16-18: "The correlations between [CH2ClI] and [CH2I2] in Bedford Basin (table 1) are consistent with this photochemical transformation" How is the correlation (CH2ClI vs CH2I2) consistent with photochemical transformation? I think that the photochemical transformation of CH2I2 to CH2ClI should decrease the correlation under the low CH2I2 production rate, and photochemical transformation should increase the correlation under the high CH2I2 production rate, which is sufficiently exceeding photodegradation of CH2I2. Authors should explain about the link between photochemical transformation and correlation (CH2ClI vs CH2I2).

4.3.1. and 4.3.2 I cannot clearly understand the difference between the seasonal variations and the sub-seasonal periodicity. Were there any periodicity within sub-seasonal

scale during summer? If there is not a periodicity, "4.3.2 sub-seasonal periodicity " is not necessary.

The authors described both "methyl iodide" and "CH3I" in the manuscript. Authors should choice either "methyl iodide" or "CH3I".

Page23 - Page 24. I cannot follow the description of the manuscript (page 23 -24) comparing with the figures. For example, from a description of "a mid-depth intrusion of salty, offshore water (as denoted by the 31 salinity contour)", I cannot read the intrusion timing comparing with temporal variation of CH3I. The authors should make a major revision in page 23 -24 adding some figures to support the explanation, as commented in "general comment".

---

## Referee Comment (RC2) · Anonymous Referee #2 · 9 Aug 2018

The manuscript describes a 3-year time series of weekly measurements of volatile iodocarbons in the cold-temperate waters of Bedford Basin. The authors use this time series, with associated data, to explore the controls on the concentrations of the three principal volatile iodocarbon compounds, to calculate air-sea flux rates of iodocarbons over the three years and to compare their results to previous time series in different regions. Directly determining the production and loss terms for these compounds and hence, being able to predictively model their emission from the oceans, is challenging and time series such as this provide useful insights into the key processes involved and key data for model validation. The 3-year duration of the study provides a unique insight into inter-annual variability of their production and emission to the atmosphere.

[Figure]

Few other studies have examined the occurrence of these compounds in the deep water of fjordic basins, as carried out in the Bedford Basin. The study is appropriate material for Ocean Science and certainly worthy of publication but requires attention to the presentation of the results and would benefit from a more focused and more succinct revision.

Concerns:

1. Manuscript format: The manuscript could be improved by more careful attention to the referencing of results. Often the reader is referred to the wrong Figure or Table. It would also be worth reviewing the sequence in which the results are presented in relation to the sequence of the text.

2. Iodide/Iodate influence. There is an overemphasis, in my view, on the link between iodide concentrations and the production of iodocarbon compounds in relation to what is presented in the paper. Although it is plausible that there may be an influence of iodide concentrations, the authors present no new information to support this, nor do they convincingly link studies that have examined iodide/iodate transformation or iodide concentrations to their own datasets. For instance, the statement made in the abstract on this issue P1L17-20 has limited bearing on what is actually presented in the paper. The same is true for the sentence in the Conclusion P27L15-17.

3. Net production of CH3I. P11L21+ It is not clear from the mass-balance calculation of net CH3I production that the mixed layer depth is taken into account? Not accounting for the mixed layer depth may have been appropriate for the study in Kiel Fjord study (Shi et al. 2014) but in the deeper waters of the Bedford Basin for the mass-balance to hold, the net production should account for the full water depth that potentially exchanges CH3I with the atmosphere. The authors should also explain why they only attempt to estimate the net production of CH3I and not the other two compounds?

4. Air-sea flux calculation: P10L15+. Previously it has been shown that accounting for the air-side resistance to air-sea transfer has an appreciable decrease in the estimated

flux of the more soluble volatile iodocarbons. For instance, the flux of CH2I2 was reduced by ∼ 30 % compared to water-side resistance-only calculations (Archer et al. 2007). The authors should state why they do not include the air-side resistance in their calculation, they may have good reason. However, it should be considered when comparing flux estimates of the different compounds and total iodocarbon flux estimates from different studies.

5. Deep-water section. Section 4.5, P21L10 to P25L2. 'Temporal variability in near-bottom water (60m)' could be usefully reduced in length in order to make it more readable and the message(s) more clear. The switch from alkylation processes that produce mono-iodinated compounds to a haloform-type process (P25L1) that is possibly linked to oxygen concentrations, is interesting but the message is lost in the complexity of the explanation of changing iodocarbon concentrations and hydrography that follows.

6. Unclear Conclusions. (Section 5) At present, the concluding section consists of a series of largely unconnected points that have been extracted from the Discussion and that lack a coherent structure. It would be more useful if the really important points were picked out and their significance described in the narrative. The easiest to read, conclusion-like comments come at the end of the section.

7. Figure and Table Legends. The figure legends and table legends could be more informative in general.

Additional points:

1. P3L5+. At some point the authors should point out that the three compounds they focus on are not the only volatile iodocarbons that are likely to be present, with minor contributions to the total from several other compounds including CH3CH2I, CH2BrI, and CHI3.

2. P7L7+: Temperature is the first environmental parameter described but does not appear in Figure 2. It would be clearer if the sequence in the text and figures corresponded. Plus, SST is not shown in either figure but is referred to in the text.

3. P7L13. No information on the seasonality of irradiance is provided until Figure 7C but it is an important part of the explanations that the authors provide for seasonal trends in both polyiodinated compounds and CH3I. I recommend removing SSS from the current Figure 2, as it is effectively repeated in Figure 3, and replacing it with irradiance information.

4. P8L1. Figure 2b shows DIN not SSS, similarly for P7L18.

5. Table 1. Please explain what the numbers actually are in the legend, Pearson's correlation coefficient? An indication of the statistical significance would also be useful.

6. Table 2. Again, more information is required in the legend that describes exactly what data is being correlated and the significance of the correlations.

7. Table 3. The legend needs rephrasing, plus more information on when theses samples were obtained would put the results in context with the seasonal study. The sequence of Tables does not match the text. The information in Table 3 is not addressed until the Discussion, after Table 4 . How was the lack of significant differences between measurements determined (P13L10)?

8. Figure 5. Please add an explanation of what the bars actually show, weekly data presumably but from single surface iodocarbon values, wind speed averaged over the week, water temperature from a single measurement during the week; these all impact on the flux value.

9. The ratio of dihalomethanes to the total volatile organic iodine is interesting but would be a stronger point if backed up with a statistical test of significance and/or an indication of the range of values between regions.

10. P15L22+ this paragraph makes some interesting points but seems to have a mixed message regarding the potential limitation of iodide concentrations on volatile iodocarbon formation rates.

[Figure]

11. P18L3-8. This paragraph describes what happened in the Kiel Fjord study, but what is the relevance to the current study. Were lagged correlations used for the Bedford Basin?

12. P19L14+. Section 4.3.2. Please clarify the points that are being made in this section, at present it is confusing.

13. P26L9+ CH3I production etc.. Of the environmental variables shown for Bedford Basin, there is a similar pattern to the temporal change of water temperature and CH3I concentrations (Table 2, Figure 7). Yet this receives little attention, despite the fact it appears to be considerably more closely related to the CH3I temporal pattern than irradiance? Why this may be the case would be worth considering.

14. P27L13. The sentence beginning 'Iodocarbon concentrations...' is difficult to interpret.

---

## Short Comment (SC1) · 13 Aug 2018

I have noted that the authors have not included all literature in their reference list that is cited in Figure 8 where contributions of iodocarbons to total organic iodine in different regions are compared. I would like to urge the authors to check all cited literature (also in Figures), and add them into the reference list (e.g. Kurihara et al., 2010, Hepach et al., 2016).

---

## Author Comment (AC1) · 28 Sep 2018

We thank the reviewer for his/her insights and very helpful comments for improving the manuscript. Please see our responses to specific comments below.

Important Point 1. Nevertheless of importance of concentration changes at 60m depth. It is difficult to follow the explanation described in the manuscript comparing with the Figure 4d (iodocarbon at 60m depth) and Figure 3 (salinity profile). It is necessary to add some detailed figures to explain timings of concentration changes of iodocarbon and water intrusions (based on analysis of salinity changes), etc.

[Figure]

Answer: We have added figure 9 (a, b and c) to clarify our discussion of the events at 60m, including intrusions, concentration increases, plateaus and "switches" (labelled 1 to 9). We also marked the 2017 mid-depth intrusion more clearly on Figure 2. We have completely rewritten this section (section 4.5) for clarity.

Important Point 2. Authors calculated the mass balance of CH3I in the top 10m of water column except for winter. However, mass balance must be calculated in the surface mixed layer defined by vertical variance of density. The thickness of the surface layer, in which water can contact with air, is crucial to estimate the balance between sea-to-air flux and net-production I the surface layer. The weekly observation of this study make it possible to set the thickness appropriately. I believe that it makes this paper more valuable.

Answer: We agree that the thickness of the layer is crucial for mass balance calculations. However, through most of the year, there is essentially little or no "mixed layer" as classically defined, as there was almost always salinity stratification between 1 and 5m. We therefore decided to perform the mass balance for the top 10m of the water column throughout the year. We did not show mass balance calculations of Pnet on Fig. 6 for the very limited periods of time when density (salinity) was constant from the surface to 10m and therefore mixing extended below 10 m.

3. Specific comments Abstract and discussion about "hypothesis that near-surface iodocarbon production is linked to reduction of iodate to iodide" I agree that dihalomethane production is supposed to associated with I2 (or HOI) production and subsequent reaction with organic matter. However, there is no evidence that I2 production is attribute to iodate reduction. Some previous studies supposed that I2 is produced as a results of iodide oxidation. Iodate reduction might occur in the surface seawater with depleting DIN. I can approve the hypothesis that near-surface iodocarbon production is linked to reduction of iodate to iodide at a discussion chapter, however, the hypothesis should not be described in the abstract.

[Figure]

Answer: We agree that we have no direct evidence about inorganic iodine cycling or speciation, and have therefore removed the hypothesis, that near-surface iodocarbon production is linked to reduction of iodate to iodide, from the abstract as suggested. We believe, however, that the timing of near-surface iodocarbon production with respect to nutrient drawdown and temperature increase can be usefully discussed in the context of experimental results in the literature (see also response to reviewer 2). Our discussion about iodine cycling in section 4.2 has been revised extensively, and we continue to mention the hypothesis in relation to results from experimental studies in section 4.3.1, and only briefly refer to it in the conclusions.

4. Abstract line 24: (2-3X) Does it mean "two to three times"? I am not sure that 2-3X is appropriate in scientific writing?

Answer: We changed 2-3X to a "factor of two" (P1L21).

5. Introduction line 6: "$CH_2ClI$ (hours) and $CH_2I_2$ (minutes)" is correct.

Answer: We corrected this (P2L22).

6. Page 7, line8: dissolved inorganic nitrogen (DIN), the definition of DIN (= $NO_3^-$ + $NO_2^-$ +$NH_4^+$) should be described at here, instead of page8 line 22.

Answer: We corrected the position of the definition of DIN. And moved it to line 6, page 8.

7. Page 8, line 6-12, etc. The authors explained degradation of organic matter using oxygen concentration. I think that AOU is better to explain the degradation of organic matter.

Answer: Yes, strictly speaking AOU is more accurate to quantify the degradation/ respiration of organic matter. But, here we feel that use of the trend/slope of oxygen concentration vs. time is sufficient and simpler to illustrate the degradation of organic matter over time. We preferred not to make this change for simplicity.
8. Page 9, line 22: "Missing from the bottom water time-series, were the ca. 1 month duration variation seen in summer time surface water", description is confused?

Answer: We added specific reference to the ca. 1 month duration peaks on line8 (page 9), and again used exactly the same wording later on in the same section, so that our discussion should now be clearer. In general we reduced our discussion of this "sub-seasonal periodicity" significantly, including no longer assigning it a separate section.

9. Page 16, line 16-18:" the correlation between [CH2ClI] and [CH2I2] in Bedford Basin (table 1) are consistent with this photochemical transformation" how is the correlation consistent with photochemical transformation? I think that the photochemical trans-formation of CH2I2 to CH2ClI should decrease the correlation under the low CH2I2 production rate, and photochemical transformation should increase the correlation un-der the high CH2I2 production rate, which is sufficiently exceeding photodegradation of CH2I2. Authors should explain about the link between photochemical transformation and correlation (CH2I2 vs CH2ClI).

Answer: We agree with the reviewer, that very efficient photochemical transformation of CH2I2 to CH2ClI should decrease the correlation between the two compounds, al-though we think that the photolysis rate may be more variable than the production rate. Where only a fraction of CH2I2 is converted to CH2ClI, we would expect a correlation. We have rewritten this section (4.2) to make this point, and note that there was no significant correlation at 1m depth, where the photolysis must be the most rapid (Page 16 and 17). We note here that Jones et al. (2010) used one-dimensional ocean mixed layer model to show that CH2ClI concentration can be almost uniform within the top 6 m, but that up to 100% of CH2I2 at 1 m depth can be photolyzed compared to at 6 m depth. In our study we also noted significant concentration differences of CH2I2 between 1 and 5 meter with the concentrations of CH2I2 at 5m being 20 times higher than at 1 m (in summer), likely due to photolysis. On the other hand, for CH2ClI, there were no significant differences between different depth (1, 5 and 10m). The strongest correlation between CH2ClI and CH2I2 occurred at 5 m depth, and this result is consistent with the photochemical transformation. We believe the discussion is now clearer on this point.

10. Section4.3.2 sub-seasonal periodicity", were there any periodicity within sub-seasonal scale during summer? If there is not a periodicity, it is not necessary.

Answer: We removed the separate section on sub-seasonal periodicity and now refer instead to "peaks of ca. 1 month duration." (P. 9 line 8).

11. Page 23-Page 24: I cannot follow the description of the manuscript comparing with figures. For example, the mid-depth intrusion of salty, offshore water (as denoted by the 31 salinity contour). I cannot read the intrusion timing comparing with temporal variation of CH3I. The authors should makes a major revision in page 23-24 adding some figures to support the explanation, as commented in "general comment"

Answer: we added a blue circle to figure 2b to highlight the mid-depth intrusion of saline water in 2017. We also added figure 9 to show changes at 60m in detail, with specific events denoted (1 through 9) and referred to in the text. We rewrote the entire discussion of the near-bottom variations for improved clarity.

---

## Author Comment (AC2) · 28 Sep 2018

We also thank reviewer #2, for the careful and useful review and suggestions for improvement. We have responded to all of the changes suggested and feel the text is now clearer and that our discussion is better grounded on the data available.

1. Manuscript format: the manuscript could be improved by more careful attention to the referencing of results. Often the reader is referred to the wrong figure or table. It would also be worth reviewing the sequence in which the results are presented in relation to the sequence of the text.

[Figure]

Answer: we have corrected and doubled checked the numbering of figures and tables. We also changed the order of figure 2 and 3, the order of figure 7 and 8.

2. Iodide/iodate influence. There is an overemphasis, in my view, on the link between iodide concentration and the production of iodocarbon compounds in relation to what is presented in the paper. Although it is plausible that there may be an influence of iodide concentrations, the authors present no new information to support this, nor do they convincingly link studies that examined iodide/iodate transformation or iodide concentration to their own datasets. For instance, the statement made in the abstract on this issuer P1L17-20 has limited bearing on what is actually presented in the paper. The same is true for the sentence in the Conclusion P27L15-17.

Answer: We removed discussion of this point from the abstract as suggested. However, following the recommendation of reviewer #1, we keep a discussion of the hypothesis within the discussion section and mention it in the conclusions. It is true that we have no data about iodide/iodate concentrations in Bedford Basin. However the relationship to the timing of DIN disappearance is notable and may relate closely to behaviour identified in laboratory studies. In general, we believe it is important to try and link findings from laboratory studies to observations in the field, which is why we attempt to connect the two lines of investigation in various sections of the discussion. This attempt reveals very clearly the need to investigate inorganic iodine cycling during these types of time-series, and our future work in Bedford Basin will attempt this so that we can test the ideas and hypotheses presented here. However we think that mentioning the hypothesis will encourage others to do the same. We have reworded some of the discussion to clarify the nature of the relations of iodide formation to organoiodine formation (section 4.2) and also to note the potential of temperature as well as nutrient drawdown to influence production rates including potential effect on reaction kinetics (section 4.3.1, P19L23).

3. Net production of CH3I: P11L21 it is not clear from the mass-balance calculation of net CH3I production that the mixed layer depth is taken into account? Not accounting

for the mixed layer depth may have been appropriate for the study in Kiel Fjord study (Shi et al. 2014) but in the deeper waters of the Bedford Basin for the mass-balance to hold, the net production should account for the full water depth that potentially exchanges CH3I with the atmosphere. The authors should also explain why the only attempt to estimate the net production of CH3I and not the other two compounds?

Answer: (see also reply to reviewer #1). We checked the mixed layer of Bedford Basin; due to strong salinity stratification, the mixed layer was shallower than 10 m almost throughout the entire time-series, except for some periods in the winter. In order to simply the calculation, we calculated the mass balance and net production for the top 10 meter water column consistently. We replotted net production rates of CH3I (Fig. 6), and no production rates are presented for periods when the layer of uniform density extended deeper than 10 m. For CH2I2 and CH2ClI, the loss due to photolysis can be very important and we did not have complete information to estimate it. So we do not show plots of net production rate of CH2I2 and CH2ClI (which we include below in this response) but rather we now give estimates of typical production rates (p. 13 line 1-4) and are careful to note the significance of various loss terms underlying our definition of Pnet for these compounds.

4. Air-sea flux calculation: P10L15. Previously it has been shown that accounting for the air-side resistance to air-sea transfer has an appreciable decrease in the estimated flux of the more soluble volatile iodocarbons. For instance, the flux of CH2I2 was reduced by ca. 30% compared to water-side resistance-only calculations (Archer et al. 2007). The authors should state why they do not include the air-side resistance in their calculation, they may have good reason. However, it should be considered when comparing flux estimates of the different compounds and total iodocarbon flux estimates from different studies.

Answer: yes, the air-side resistance should be considered during the calculation of the flux. We recalculated fluxes including air-side resistance, and the reduction of the flux of CH2I2 averaged 24%, 10% for CH2ClI and 2% for CH3I. The revised text is

presented in section 3.3. We also were careful to adjust the flux estimates of Shimizu et al. (2017), prior to comparing with our estimates and those of Archer et al. (2007). The adjustments are significant and, interestingly, lead to closer correspondence of the annual air-sea fluxes of Iorg from the three environments (see Table 5).

5. Deep-water section. Section 4.5, P21 L10 to P25 L2. ' temporal variability in near bottom water (60m)' could be usefully reduced in length in order to make it more readable and the messages more clear. The switch from alkylation processes that produce mono-iodinated compound to a haloform-type process (P25L1) that is possible linked to oxygen concentrations, is interesting but the message is lost in the complexity of the explanation of changing iodocarbon concentration and hydrography that follows.

Answer: In section 4.5 ' temporal variability in near bottom water (60m)', we added figure 9 to help clarify the discussion. Special events labelled 1 to 9 present the timing of switching from alkylation processes to a haloform-type reaction, and of intrusions. We completely rewrote the description of these events and while still substantial, the text has been shortened considerably and is, we feel, much easier to follow.

6. Unclear conclusions. (Section 5) At present, the concluding section consists of a series of largely unconnected points that have been extracted from the discussion and that lack a coherent structure. It would be more useful if the really important points were picked out and their significance described in the narrative. The easiest to read, conclusion-like comments come at the end of the section.

Answer: We restructured and shortened the conclusion section, dispensing with some minor points, and emphasising the more general conclusions towards the end.

7. Figure and Table legends. The figure legends and table legends could be more informative in general.

Answer: We added more information to the captions.

Additional points:

1. P3L5, At some point the authors should point out that the three compounds they focus on are not the only volatile iodocarbons that are likely to be present, with minor contributions to the total from several other compounds including CH3CH2I, CH2BrI and CHI3.

Answer: yes, VOIs also include CH3CH2I, CH2BrI and CHI3, although they are usually minor contributions compared with other three compounds. We added this to line 2 of page 3.

2. P7L7: Temperature is the first environmental parameter described but does not appear in Figure 2. It would be clearer if the sequence in the text and figures corresponded. Plus, SST is not shown in either figure but is referred to in the text.

Answer: We changed the sequence of figures. The new figure 2 is about seasonal patterns of environmental and biological variables in Bedford Basin, and temperature is shown in fig 2a. And SST is corrected to temperature in the text (P7L23).

3. P7L13: No information on the seasonality of irradiance is provided until figure 7c, but it is an important part of the explanations that the author provides for seasonal trends in both polyiodinated compounds and CH3I. I recommend removing SSS from the current figure 2, as it is effectively repeated in figure 3, and replacing it with irradiance information.

Answer: We added seasonality of irradiance in figure 3f. For sea surface salinity we refer to this now as "salinity" in section 3.1, so we kept the figure (figure 3c). These data are especially important for showing the near-surface stratification.

4. P8L1: Figure 2b shows DIN not SSS, similarly for P&L18.

Answer: We corrected this in the text (see P7L9).

5. Table 1: Please explain what the numbers actually are in the legend, Pearson's correlation coefficient? An indication of the statistical significance would also be useful.

Answer: They are the R2 values (Pearson's correlation coefficient) while p is < 0.05. We added more information in the caption.

6. Table 2: Again, more information required in the legend that describes exactly what data is being correlated and the significance of the correlations.

Answer: They are the R2 values (Pearson's correlation coefficient) while p is < 0.05. We added more information in the caption.

7. Table 3: The legend needs rephrasing, plus more information on when these samples were obtained would put the results in context with the seasonal study. The sequence of tables does not match the text. The information in table 3 is not addressed until the discussion, after table 4. How was the lack of significant differences between measurements determined (P13L10)?

Answer: We swapped the order of Table 3 and 4 and added more information in the caption of Table 4. The goal of these measurements was to look for potential influence of nearshore, macroalgal production. The highest concentration of $CH_2I_2$ (18.6 pmol/L). We added a sentence to section 4.1 (P13L18), which compares the nearshore measurements to the mean and standard deviation of measurements made at the central Basin sampling site during the month of July. Compared to the very significant differences of bromocarbons measured close to or far from macroalgal beds (e.g. Klick, 1992), the difference of iodocarbons was certainly not significant.

8. Figure 5: Please add an explanation of what the bars actually show, weekly data presumably but from single surface iodocarbon values, wind speed averaged over the week, water temperature from a single measurement during the week, these all impact on the flux value.

Answer: This is a useful point. We added more information in the caption and also in the text (section 3.3) to explain how we calculated the fluxes and what the averages represent. We recalculated the fluxes so that the bars in figure 5 represent weekly

averages of daily flux estimates (based on daily average wind speeds and interpolated daily values based on weekly water sampling). This is now explained clearly in the text.

9. The ratio of dihalomethanes to the total volatile organic iodine is interesting but would a stronger point if backed up with a statistical test of significance and/ or an indication of the range of values between regions.

Answer: We assume here that the reviewer was referring to the results shown in Fig. 7. We have added a little more text on the comparison between regions to the text in section 4.2 on both pages 14 and 15. Basically a 1-way ANOVA showed no significant difference between means for the three regions for either total organoiodine content (Iorg) or for the percentage of the total given by different compounds. On the other hand, if we pooled results into only two regions (coastal and shelf+open ocean) then a t-test showed a significant difference for Iorg at the 95% confidence level. Although Fig. 7 appears to reveal differences in the relative contributions of different compounds between regions, we now state clearly that the overall differences are not statistically significant.

10. P15L22: this paragraph makes some interesting points but seems to have a mixed message regarding the potential limitation of iodide concentrations on volatile iodocarbons formation rates.

Answer: We rewrote this paragraph to clarify the description and we hope the arguments about potential for inverse correlations of iodocarbons with iodide are now clearer.

11. P18L3-8: this paragraph describes what happened in the Kiel Fjord study, but what is the relevance to the current study. Were lagged correlations used for the Bedford Basin. Answer: We did not use lagged correlation analysis for the Bedford Basin as the end-result of the earlier study in the Kiel Fjord was that detailed correlation analysis may have reached its practical limit of utility for identifying causation. This paragraph was rewritten (line 1-7, page18) to make this point. 12. P19L14: Section 4.3.2. Please

clarify the points that are being made in this section, at present it is confusing.

Answer: we removed this separate section and now simply refer to maxima of ca. 1 month duration.

13. P26L9: CH3I production etc. of the environmental variables shown for Bedford Basin, there is a similar pattern to the temporal change of water temperature and CH3I concentration (Table 2, figure 7). Yet this receives little attention, despite the fact it appears to be considerably more closely related to the CH3I temporal pattern than irradiance? Why this may be the case would be worth considering.

Answer: We changed the sequence of Fig. 7 and 8, and added a reference to Fig. 7a in section 4.3.1. We had already discussed the temperature influence in Page 19 line 23, and emphasise the correlation between CH3I production and temperature.

14. P27L13: The sentence beginning 'iodocarbon concentrations….' is difficult to interpret.

Answer: We rewrote the whole conclusion and removed this sentence from the conclusion.

[Figure]

[Figure]

**Fig. 1.** Variation of the net production rate of VOIs in the upper 10 m from 2015 to 2017. No data are plotted when the layer of uniform density extended below 10 m.

---

## Author Comment (AC3) · 28 Sep 2018

We now add the missing references into our list. We have also doubled checked the reference list to make sure it's complete.

---

## Author Response (AR2)

We thank editor for the careful and useful suggestions for improvement. We have responded to all of the changes suggested.

1. *In this paper new measured data from DIN, dissolved oxygen and chlorophyll a are presented besides the iodocarbons. For these measurements the reader is referred to Burt et al. (2013). However, the data from Burt et al were collected before the data from the present study. I think it would be useful and necessary to present the additional measurements briefly including their precision/accuracy.*

Answerer: We have added the additional introduction of measurements including their precision in P5L22.

2. *P7, L13 psu is not a unit. I would advise not to use it.*

Answerer: We have deleted psu from text and also corrected this in Fig. 3.

3. *P14, L12 "is shown in F." There appears to be something missing here.*

Answerer: We have corrected "is shown to F" to "is shown in Fig.7" in text P14L19.

4. *P17, L16 Please define DOM here as this has not been done before.*

Answerer: We corrected DOM to dissolved organic matter (DOM)" in text P18L1.

5. *P26-27, L23-1 delete "although both factors,"*

Answerer: We deleted these words from text.

6. *References:*

*P30, L22 224Ra with 224 superscript*

*P33, L3 CH2I2, CH2ICl, and CH2IBr please use subscripts*

*P33, L5 CH3I, C2H5I, 1-C3H7I, and 2-C3H7I please use subscripts*

*P34, L22 Please add doi for completion of reference.*

*P35, L19 CH 2 Br 2, CH 2 I 2 and CH 2 BrI please use subscripts, and check format*

*P36, L6 delete: [online] Available from:*

*P36, L13 ($CH_3I$) please use subscripts*

*P36, L14 Correct author names: Rattigan, O.V., Shallcross, D.E., and R. Anthony Cox, R.A.:*

*P36, L15 $CF_3I$, $CH_3I$, $C_2H_5I$ and $CH_2ICl$ please use subscripts, and check format*

*P37, L21 delete: (BG)*

*P38, L1 delete: (BG)*

*P38, L11 This reference format needs a doi*

*P38, L14 This reference format needs a doi*

Answerer: We have corrected the formation all mentioned references.

We thank the reviewer again to improve the manuscript. Please see our responses to the specific comments below.

1. *Page2 line8: Does 'VSLS' mean 'very short-lived source gass' ? WMO report (ozone depletion) refers 'VSLS' as 'very short-lived halogenated substances'.*

Answerer: We have corrected VSLS to very short-lived halogenated substances in P2L8.

2. *Page7, line13: Unit of salinity should be unified through the manuscript and figures. i.e. 'psu' or 'non-unit' .*

   Answerer: We have deleted psu from text and also corrected this in Fig. 3.

3. *Page9, line23: strongest correlation (R = 0.7), please add 'R = '.*

Answerer: We have added "R=0.7" in the text P10L9.

4. *Page24, line10-15, and line16-: Authors say that event 7 is "switch" in line10-15, and say that event 7 is reflected intrusion of saltier water. Is event 7 a mixture of "switch" and "intrusion of saltier water" ? Please explain clearly.*

Answerer: Event 7 is "switch", at the same time a intrusion had happed in the mid-layer water (30m deep). It cannot be ruled out that mixing with this intrusion contributed in some way to the near-bottom increase of $CH_2I_2$. We have rewritten this paragraph (P25L1-P25L15) for clarity.

5. *Page22, line1-2: Please check parenthesis. (e.g. Moore ---Yamamoto et al., 2001) is correct.*

Answerer: We have corrected the brackets in P22L9.

6. *Page21, line 16: Is 'between' OK? Please check 'between" or 'among'.*

Answerer: This is a tricky grammatical distinction that made us think and do some research, however we have decided that "between" is correct. There is quite a good treatment here:

https://blog.oxforddictionaries.com/2015/06/29/grammar-myths-among-or-between/

where it is noted that the common advice to use "between" for two things and "among" for more than two is:

"now regarded as outdated and out of step with current usage. In fact, as the Oxford English Dictionary (OED) states, 'In all senses, between has been, from its earliest appearance, extended to more than two': there's an example of this from the year 971 (yes, not 1971!). Contemporary authorities (such as Pocket Fowler's Modern English Usage) advise that it's perfectly acceptable to use between or among in certain contexts when referring to more than two participants:

*He divided his fortune between his four children.*

*He divided his fortune among his four children."*

Later on it is stated:

"*Between* is preferred when we talk about a relationship of difference, no matter how many people or things are involved:

*The difference between those results is not statistically significant.*

*X  The difference among those results is not statistically significant.*"

In our situation we are actually referring to a difference (or lack thereof) between three depths, so I think it is correct to use "between".

7. *Page27: Why is the production rate of $CH_3I$ in the Kiel Fjord by monthly average expected to be smaller? Why is weekly incubation experiment expected to be closed to the value of this study ? Are these associated to microbial degradation of $CH_3I$? Authors should explain clearly.*

Answerer: The maximum production rates from the Kiel Fjord study were smaller as they were based on monthly average (and therefore "smoothed") concentrations. Shi et al. (2014a) also conducted weekly incubation experiments which gave *in vitro* values of $P_{net}$ which were closely comparable with the field-based estimates in Kiel Fjord. Evidence for a poorly characterized loss process, possibly microbial degradation, was in fact observed in the Kiel Fjord incubation experiments (Shi et al., 2014a). We have clarified them in section 4.7 (P27).